# Quantum spin resonance in engineered proteins for multimodal sensing

Gabriel Abrahams[1✉], Ana Štuhec[2,8], Vincent Spreng[1,3,8], Robin Henry[1], Idris Kempf[1], Jessica James[1], Kirill Sechkar[1], Scott Stacey[1], Vicente Trelles-Fernandez[1], Lewis M. Antill[2,4], Christiane R. Timmel[2], Jack J. Miller[5], Maria Ingaramo[6], Andrew G. York[6], Jean-Philippe Tetienne[7] & Harrison Steel[1✉]

Sensing technologies that exploit quantum phenomena for measurement are finding increasing applications across materials, physical and biological sciences[1–7]. Until recently, biological candidates for quantum sensors were limited to in vitro systems, had poor sensitivity and were prone to light-induced degradation. These limitations impeded practical biotechnological applications, and high-throughput study that would facilitate their engineering and optimization. We recently developed a class of magneto-sensitive fluorescent proteins including MagLOV, which overcomes many of these challenges[8]. Here we show that through directed evolution, it is possible to engineer these proteins to alter the properties of their response to magnetic fields and radio frequencies. We find that MagLOV exhibits optically detected magnetic resonance in living bacterial cells at room temperature, at sufficiently high signal-to-noise for single-cell detection. These effects are explained through the radical-pair mechanism, which involves the protein backbone and a bound flavin cofactor. Using optically detected magnetic resonance and fluorescence magnetic-field effects, we explore a range of applications, including spatial localization of fluorescence signals using gradient fields (that is, magnetic resonance imaging using a genetically encoded probe), sensing of the molecular microenvironment, multiplexing of bio-imaging and lock-in detection, mitigating typical biological imaging challenges such as light scattering and autofluorescence. Taken together, our results represent a suite of sensing modalities for engineered biological systems, based on and designed around understanding the quantum-mechanical properties of magneto-sensitive fluorescent proteins.

Coupling electromagnetic fields with biological processes through fluorescence has revolutionized quantitative biology[9]. Meanwhile, quantum-sensing tools (that is, those for which function arises from electron-spin- or nuclear-spin-dependent processes) have been developed for their unique advantages in biological applications, but have until recently been limited to realizations using non-biological probes[1–3] or measurements under ex vivo conditions[4–6,10]. We previously reported the development of a library of magneto-responsive fluorescent protein (MFP) variants derived from the LOV2 domain (broadly termed MagLOV), which exhibit fluorescence signals with large magnetic-field effects (MFEs)[8] (Fig. 1a). With this work, we demonstrate that at room temperature, we can detect optically detected magnetic resonance (ODMR)[11,12] in living cells expressing these fluorescent proteins, including at the single-cell level. The ODMR signature implies a quantum system with properties and dynamics influenced by the local environment, opening up a broad range of possibilities for cellular biosensing[1]. The ODMR arises from electron spin resonance (ESR) that we propose originates from a spin-correlated radical pair (SCRP)[11,13] (Fig. 1b) involving the LOV2 domain non-covalently bound flavin cofactor chromophore. This theory is based on previous evidence[14] and supported by the MFE, ODMR and spectroscopic studies we present.

Both MFE and ODMR signals are straightforward to detect in cells on a standard wide-field fluorescence microscope, supporting further development and application of this discovery. Beyond the ease of detecting MFPs' magnetic resonance via emission, MFPs are advantageous over other candidate spin sensors for biological uses because they can be expressed directly in the host organism, allowing direct coupling and regulation by biological processes, and because their performance can be engineered genetically, such as through rational design or directed evolution. This engineerability is demonstrated through a selection approach previously reported[8] and here used to generate protein variants specialized for sensing applications.

We demonstrate applications of MFPs as reporters that can be used for lock-in signal amplification in noisy measurement environments

[1]Department of Engineering Science, University of Oxford, Oxford, UK. [2]Department of Chemistry, University of Oxford, Oxford, UK. [3]Institute of Pharmacy and Molecular Biotechnology (IPMB), Faculty of Engineering Sciences, Heidelberg University, Heidelberg, Germany. [4]Institute of Quantum Biophysics, Department of Biophysics, Sungkyunkwan University, Suwon, Republic of Korea. [5]The MR Research Centre, Aarhus University, Aarhus, Denmark. [6]Calico Life Sciences, South San Francisco, CA, USA. [7]Department of Physics, School of Science, RMIT University, Melbourne, Victoria, Australia. [8]These authors contributed equally: Ana Štuhec, Vincent Spreng. ✉e-mail: gabriel.abrahams@eng.ox.ac.uk; harrison.steel@eng.ox.ac.uk

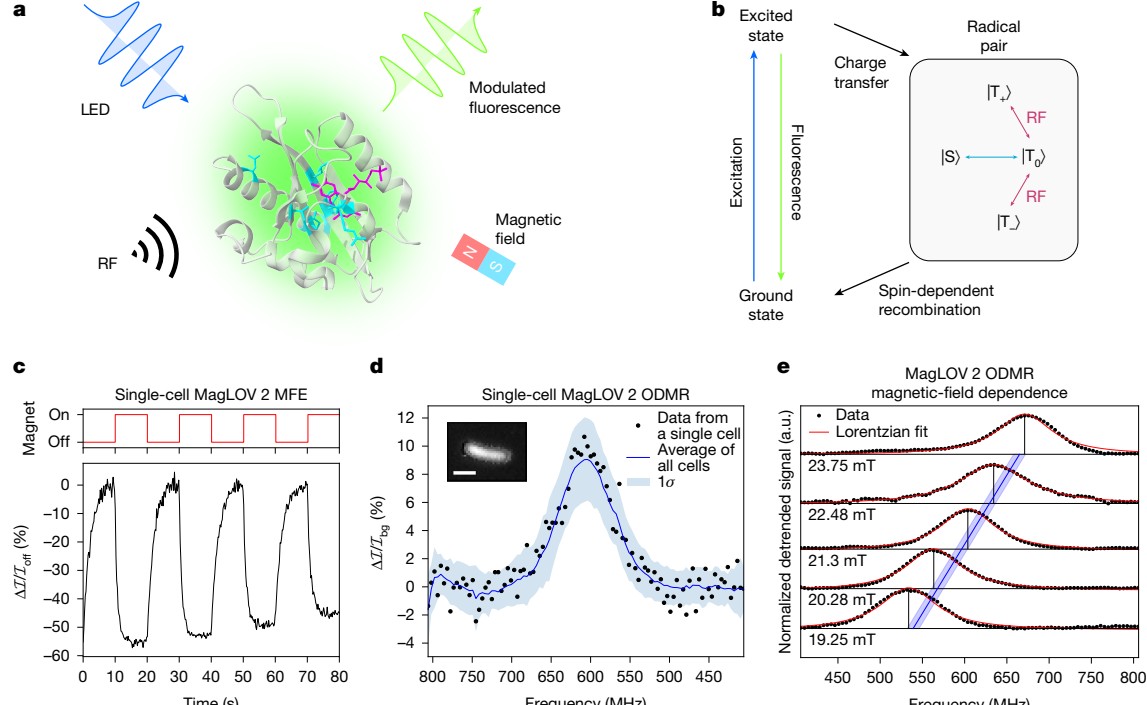

**Fig. 1 | MFE and ODMR in MFPs. a**, Structure of AsLOV2 PDB 2V1A[58] with MagLOV 2 mutations highlighted. Spin transitions driven by radio-frequency (RF) fields in the presence of a static magnetic field are optically detected via fluorescence measurements. LED, light-emitting diode. **b**, Simplified photocycle diagram in the case of a large external magnetic field. The radical pair is born in a triplet state $|T\rangle$ with spin projections $|T_0\rangle$, $|T_+\rangle$ and $|T_-\rangle$ (refs. 20,21) and undergoes field-dependent singlet–triplet interconversion to the singlet state, $|S\rangle$, driven by nuclear-electron spin–spin interactions. **c**, Microscope measurement of a single cell expressing MagLOV 2 showing an MFE of about 50%. For MFE measurements, the magnetic field was switched between 0 mT and 10 mT, here with a period of 20 seconds. The intensity over time is integrated over pixels covering the single cell, with a background photobleaching trendline removed. MFE is calculated as $(\mathcal{I} - \mathcal{I}_{off})/\mathcal{I}_{off} = \Delta\mathcal{I}/\mathcal{I}_{off}$. **d**, Data from a single cell expressing MagLOV 2 shows an ODMR signal with about 10% contrast. The static field $B_0$ is about 21.6 mT. The blue line (shading) is the mean (standard deviation) of all single-cell data in a field of view (about 1,000 cells). Inset: microscope image cropped to a single cell expressing MagLOV 2. Scale bar, 2 μm. **e**, The static magnetic field $B_0$ was varied by adjusting the magnet's position, and the ODMR spectra recorded. The blue line shows the theoretical prediction for the resonance frequency of an electron spin with gyromagnetic ratio $\bar{\gamma}_e = 28$ MHz mT$^{-1}$ or $g_e = 2.00$, with the shaded region representing uncertainty ($\pm 0.25$ mT) in the magnetic-field strength (as determined by a Hall probe) at the sample position.

(as often encountered in biological applications[15]), and to enable signal multiplexing by engineering variants with differing dynamic responses. We also show that the MagLOV MFE is attenuated by interaction with magnetic resonance imaging (MRI) contrast agents, with a dose-dependent effect consistent with spin relaxation, implying MagLOV's ability to sense the surrounding environment. Finally, we realize applications to spatial imaging; because the ODMR resonance condition depends on the static magnetic field at the location of the protein, it is possible to use gradient fields (as in MRI) to determine the spatial distribution of MFPs with scattering-independent measurements of a sample's fluorescence. We demonstrate this by building a fluorescence MRI instrument based on a small-animal MRI coil with a one-dimensional magnetic gradient, which we apply to simultaneously localize the depth position of multiple bands of bacterial cells embedded in a three-dimensional volume. Ultimately, this work represents a proof of principle for MFPs and their applications, which may develop into a paradigm of quantum-based tools for biological sensing, measurement and actuation.

## ODMR in living cells

Reaction yield detected magnetic resonance (RYDMR; a form of ODMR) is both a diagnostic test for the existence of a proposed SCRP[16] and, owing to its relative simplicity, an effective measurement modality for performing readout from quantum-sensing devices in biological and materials applications[1,2,17–19]. ODMR studies the spin dynamics of such systems through the application of oscillating radio-frequency

magnetic fields, $B_1$, resonant with spin transition energies, and facilitates optical readouts through light emission or absorption. SCRPs are transient reaction intermediates, often generated by (photoinitiated) rapid electron transfer from a donor (in biological systems, often an aromatic amino acid) to an acceptor—here flavin mononucleotide (FMN), which serves as the field-sensitive fluorophore in our system. As electron transfer occurs under conservation of total spin angular momentum, the total spin of the radical pair is defined by that of its molecular precursor (either a singlet $|S\rangle$ ($S = 0$) or a triplet state $|T\rangle$ ($S = 1$), where $S$ is the total spin quantum number). If the radicals in the pair are weakly coupled, the spin system is created in a superposition of the uncoupled states. Consequently coherent interconversion occurs between the $|S\rangle$ and $|T\rangle$ states, driven by the interactions between electron and nuclear spins (such as $^1$H and $^{14}$N; Fig. 1b). At zero and low static magnetic fields, this interconversion occurs rapidly involving all four states, but at higher fields, singlet–triplet mixing is restricted to $|S\rangle$ and $|T_0\rangle$ with $|T_\pm\rangle$ energetically isolated from $|S\rangle$ and $|T_0\rangle$ by Zeeman splitting[20,21]. Importantly, singlet and triplet radical pairs have different fates: whereas the singlet pair is able to recombine to yield the ground-state donor and acceptor, the triplet cannot do so and instead forms other forward photoproducts (for example, by protonation or deprotonation), a route also open to the singlet pair. Under continuous illumination, the impact of the field on the singlet–triplet interconversion can be detected conveniently if either donor or acceptor or both form fluorescent excited states. In this case, a drop in fluorescence with an applied strong (about 10 mT) magnetic field ($B_0$) is expected for a triplet-born pair as mixing into the recombining singlet state is

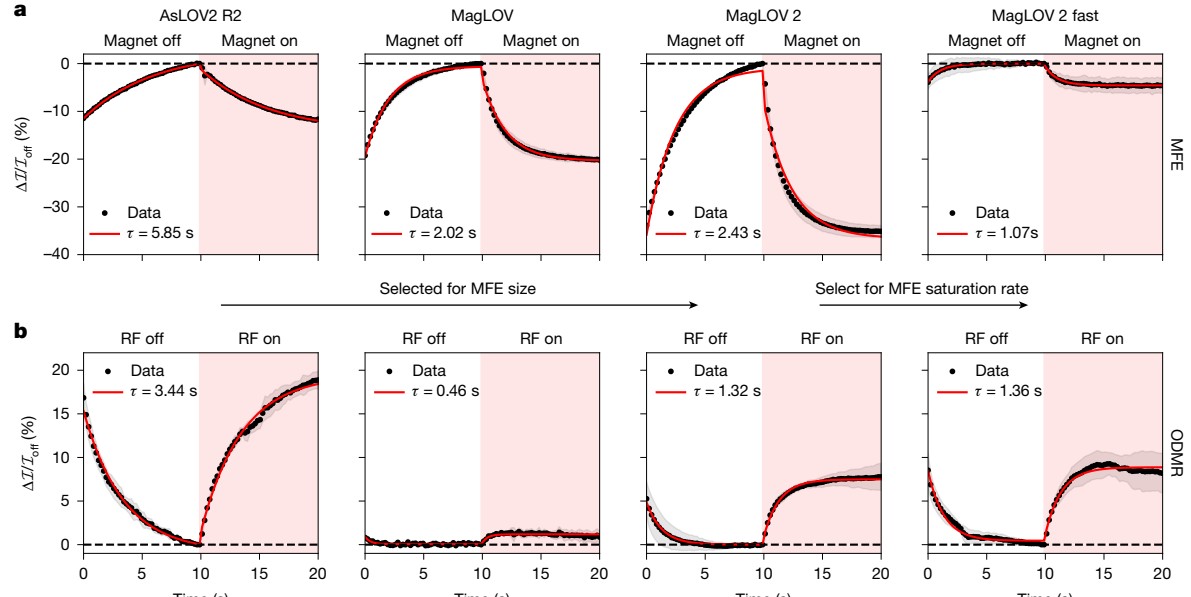

**Fig. 2 | Engineering of MFE and ODMR dynamics. a**, MFP variants were engineered by mutagenesis and directed evolution; from AsLOV2 R2 to MagLOV 2 selection was performed to increase the MFE magnitude at saturation, whereas MagLOV 2 fast was selected for increased rate (that is, reduced time constant $\tau$ when the MFE with time $t$ is modelled as $e^{-t/\tau}$ (ref. 25), shown in red fit). In these measurements, a magnetic field of $B_0 = 10$ mT was switched on and off with a period of 20 seconds. The traces shown are the average (and standard deviation, shaded) over multiple periods. As a result, part of the variability results from the background photobleaching curve fit which attempts to match the true background curve (see Supplementary Note 9 for details). **b**, A similar experiment was performed using ODMR on-resonance, with a constantly applied static field of $B_0 = 21.6$ mT and corresponding resonant field $B_1$ frequency of $\omega_{RF} = 604$ MHz switched on and off with a 20-second period, averaged as above. Both sets of experiments were performed in identical imaging set-ups with an illumination intensity of about 800 mW cm$^{-2}$ at 450 nm. Note that $B_0$ in **a** is half that of $B_0$ in **b**, explaining why the ODMR contrast is in some instances larger than the MFE.

impeded; hence, the system exhibits an MFE. Meanwhile, application of an additional resonant radio-frequency field ($B_1$) would reconnect $|T_0\rangle$ with $|T_\pm\rangle$, leading to an increase in fluorescence intensity, measured as ODMR.

This hypothesis motivated consideration of MFE and ODMR measurements as approaches to confirm that the radical pair mechanism of MagLOV is triplet-born[20]. Previous studies have similarly explained MFEs arising from interactions between a flavin cofactor and protein in terms of the SCRP mechanism[22–28], with the most prominent example being cryptochromes implicated in avian magnetoreception[29–32]. Furthermore, radical-pair intermediates are known to form in the LOV2 domain *Avena sativa* phototropin 1 (AsLOV2) variant C$_{450}$A (the precursor protein to MagLOV)[33], as well as in related LOV domains[34–36]. Parallel with our results, ODMR has recently been reported in purified enhanced yellow fluorescent protein (EYFP) at room temperature, and in mammalian cells at cryogenic temperature[10], and subsequent preprints have reported ODMR in purified protein solutions of *Drosophila melanogaster* cryptochrome (*Dm*Cry), mScarlet-flavin and MagLOV at room temperature, as well as mScarlet-flavin in *Caenorhabditis elegans* at room temperature[32,37].

First, we measured the MagLOV MFE (see example trace in Fig. 1c) using a custom-built fluorescence microscopy platform, confirming that the variant MagLOV 2 exhibits a large MFE of $\Delta\mathcal{I}/\mathcal{I}_{off} = -50\%$, where $\mathcal{I}_{on,off}$ is the fluorescence intensity when the electromagnet is on or off, and $\Delta\mathcal{I} = \mathcal{I}_{off} - \mathcal{I}_{on}$. In Fig. 1d, we show the ODMR resonance of a single cell and the ODMR resonance averaged over many (about 1,000) cells in a field of view. As expected, the signal-to-noise is significantly improved by averaging over many cells; however, it is also possible to extract an ODMR signal from a single cell, with an ODMR contrast of 10% (Fig. 1d). The remarkable per-cell magnetic sensitivity of $\eta_0 = 26$ µT Hz$^{-1/2}$ (Supplementary Note 1) is afforded by a combination of optical detection and the high spin polarization of the radical-pair system[1]. Next, we recorded ODMR spectra at various static

($B_0$) fields by adjusting the $z$ position of the static magnet (Fig. 1e). The central ODMR resonance follows the expected ESR relationship $f_{RF} = \bar{\gamma}_e B_0$ (with $\bar{\gamma}_e$ the electron gyromagnetic ratio), confirming that the radio-frequency field is driving spin transitions of a spin-1/2 electron. Finally, we performed additional control experiments confirming the ODMR signal's source, verifying against negative controls that an ODMR signal is present only when the MagLOV protein is present (Supplementary Note 2).

## Engineering of MFEs

The AsLOV2 domain has been widely used as a starting point for engineering optogenetic and other light-dependent protein functionalities[38]. For MFPs, depending on a target application (such as lock-in signal detection, multiplexing or sensing), one could choose to optimize for metrics including MFE size, rate of MFE saturation, ODMR contrast, ODMR saturation rate and others. Here we demonstrate this potential by performing selection to improve MFE contrast and saturation rate, generating variants specialized for applications demonstrated later in our work.

Starting from ancestor variant AsLOV2 C$_{450}$A (refs. 8,39), we used directed evolution to create variants of MFPs (summarized in Supplementary Note 3). This engineering process involved successive rounds of mutagenesis (introducing all single amino acid changes to a given variant), followed by screening of samples from this variant library to select for increased MFE magnitude, eventually yielding MagLOV 2. To demonstrate the possibility of selecting on another metric using the same methodology, we further engineered MagLOV 2 by selecting for maximization of saturation rate (rather than magnitude) of MFE, producing 'MagLOV 2 fast'. We chose four variants to characterize in detail, which was done both using single-cell microscopy (Fig. 2a) and measurement of bulk cell suspensions (Supplementary Note 4).

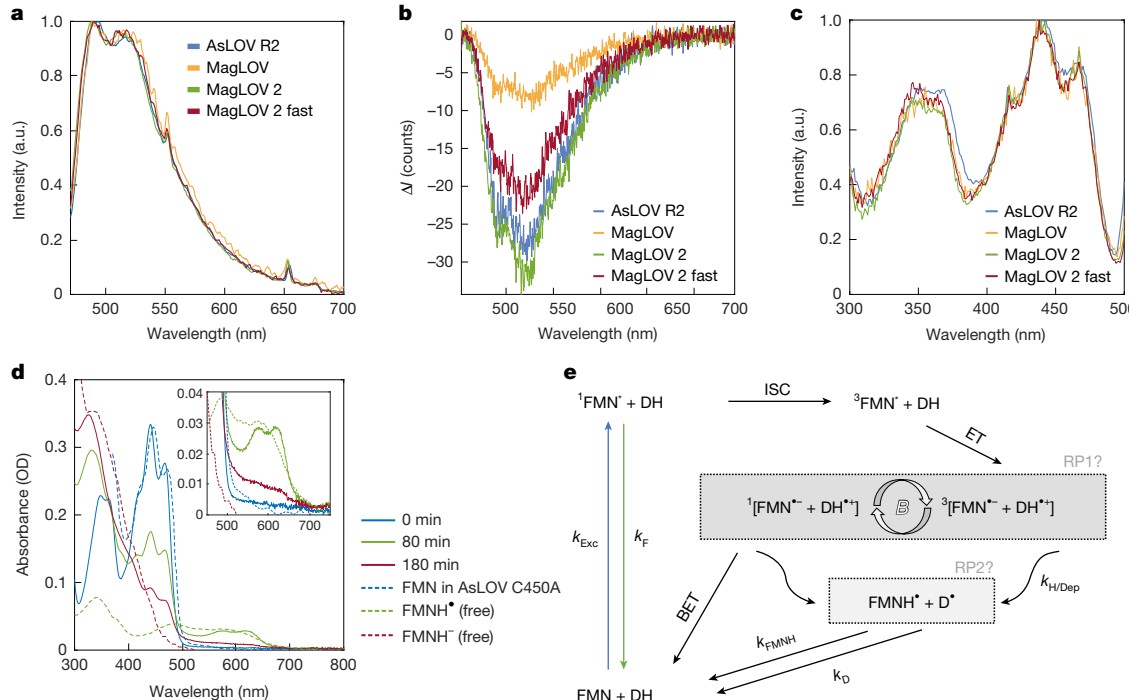

**Fig. 3 | Spectroscopic characterization evidences bound flavin-based radical pair. a**, Normalized emission spectra with 450-nm excitation acquired for cell suspensions in PBS buffer. The spectra have been smoothed with a moving average filter of 1-nm bandwidth. **b**, Wavelength dependence of magnetic-field-induced change of emission intensity where $\Delta\mathcal{I} = \mathcal{I}(B_0 = 10\text{ mT}) - \mathcal{I}(B_0 = 0\text{ mT})$. It is noted that here we display the absolute intensity change as division by $\mathcal{I}(B_0 = 0\text{ mT})$ would obscure the wavelength dependence. A moving average filter of 1-nm bandwidth was applied to the spectra. **c**, Normalized excitation spectra for 510-nm emission for bulk MagLOV cell suspensions in phosphate-buffered saline (PBS) buffer. Vibrational fine structure (multiple peaks) in the $S_0 \rightarrow S_1$ and $S_0 \rightarrow S_2$ bands (where $S_n$ is the $n$th excited singlet state) centred at about 450 nm and about 350 nm, respectively, indicates that the emitting flavin is bound. **d**, Ultraviolet–visible absorption spectrum of purified MagLOV 2 fast at different

times after the onset of blue LED illumination. Inset: a reduced wavelength range with the characteristic absorption of FMNH$^{\bullet}$. Literature reference spectra are shown as dashed lines[41,59,60]. **e**, Proposed photoscheme. Following photoexcitation ($k_{Exc}$), the excited singlet flavin ($^1$FMN*) can either emit a photon ($k_F$) or undergo intersystem crossing (ISC) to the excited triplet state ($^3$FMN*). The primary radical pair (RP1) is formed by electron transfer (ET) from a nearby donor, and can undergo singlet–triplet interconversion, which is altered in the presence of an applied magnetic field ($B$). Only the overall singlet RP can undergo back-electron transfer (BET) to reform the ground state, whereas either RP can form secondary (perhaps spin uncorrelated) radicals (RP2) through protonation and/or deprotonation reactions ($k_{H/Dep}$). These long-lived secondary radicals return to the ground through slow redox reactions ($k_{FMNH}$, $k_D$).

The observed differences in MFE can be interpreted based on past work investigating flavin magnetic-field-sensitive photochemistry: MFE enhancement kinetics (that is, time to MFE saturation) are determined by the ratio between the rates that donor and acceptor free radicals return to the ground state[25]. With the mutations introduced in MagLOV 2 fast, the time constant of the response is decreased, which may indicate that the acceptor return rate increases relatively to the donor. Interestingly, we observe the ODMR contrast and rates of each variant differ significantly (Fig. 2b), but not necessarily in simple correlation with MFE magnitude. This raises the possibility of engineering orthogonal fluorescence signatures, expanding again the number of tags available for multiplexing. For instance, with further engineering the total might be (number of emission colours) × (number of resolvable MFE signatures) × (number of resolvable ODMR signatures).

## Bound flavin and radical-pair mechanism

Adopting the SCRP model prompts consideration of the electron donor and acceptor identities. Both previous studies[33,35] and our spectral data (Fig. 3) support the identification of the acceptor molecule as the FMN cofactor. For all variants of MFPs expressed in cells, we find that the wavelength-resolved fluorescence intensity modulated by the applied magnetic field ($\Delta\mathcal{I}$) (Fig. 3a,b) matches the FMN emission spectrum[39,40], supporting that both MFE and ODMR are detected on the flavin emission. The excitation spectrum shows vibrational fine structure (Fig. 3c) and is in excellent agreement with the dark-state

absorption spectrum of AsLOV2 C$_{450}$A (ref. 41), confirming that the emission originates from bound FMN. Furthermore, this corroborates control experiments (Supplementary Note 2) that show that observed ODMR signatures are not a result of cellular autofluorescence. The absorption spectrum of purified MagLOV 2 fast, before and after continuous irradiation with blue light, is shown in Fig. 3d (the full temporal evolution is provided in Supplementary Note 4). The slow formation of the stable radical FMNH$^{\bullet}$ after several minutes of illumination is characterized by the appearance of a broad band featuring 2 peaks centred around 575 nm and 615 nm, as observed for AsLOV C$_{450}$A, and accompanied by the expected decrease in ground-state absorption, centred around 450 nm (ref. 41). At later times, most of the flavin is converted to the fully reduced form, FMNH$^-$, resulting in a rise in absorption around 325 nm, although some remains present as FMN and FMNH$^{\bullet}$. The blueshift in MagLOV-bound FMN absorption relative to AsLOV C$_{450}$A, evident both in the excitation and absorption spectra, may indicate a change in polarity of the flavin binding pocket. The observed photostability of MagLOV correlates with the slow formation of FMNH$^{\bullet}$ on a timescale of minutes in these conditions. In comparison, related flavoproteins, such as cryptochrome, are reduced to the semiquinone state within seconds, even under much weaker irradiation intensities[42]. Unlike the recently reported MFE in mScarlet3 and FMN mixtures, which rely on a bimolecular reaction between the excited-state mScarlet3 and a fully reduced flavin in solution[28], MagLOV requires no pre-illumination or additives for an MFE to develop as the FMN is non-covalently bound.

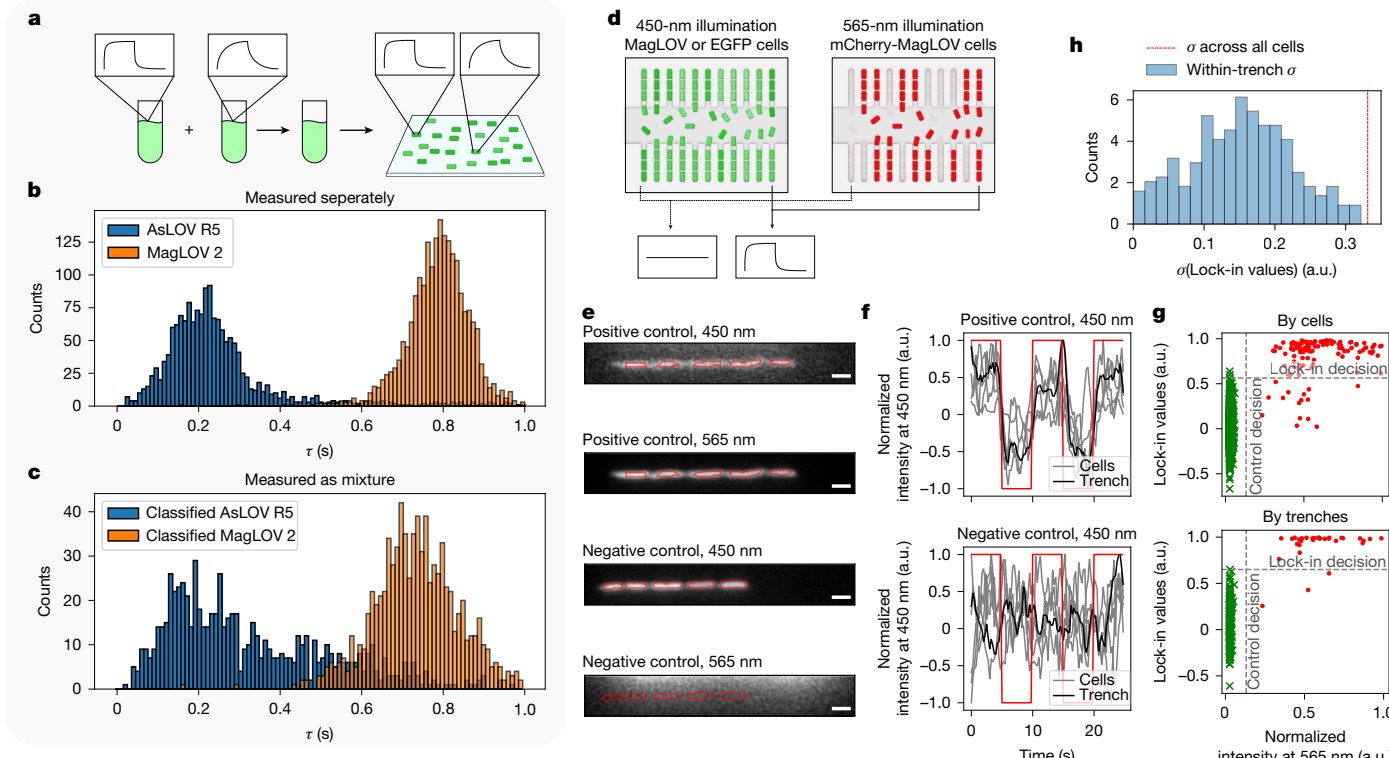

**Fig. 4 | Multiplexing and lock-in applications using MFE. a**, MagLOV variants can be used to label cell populations, which can be identified when mixed based on differing MFE responses. **b**, Exponential curves with timescale parameter $\tau$ were fit to the MFE of each cell in each field of view. Here populations are measured separately, illustrating that MagLOV 2 has a greater MFE saturation timescale than AsLOV R5. **c**, Populations were mixed in an equal ratio and a classifier (Methods) trained on the time-series data in **b** was used to classify the mixture into two subpopulations. **d**, Schematic of the microfluidic chip, composed of single-cell-wide trenches (vertical channels). Cells were engineered to weakly express MagLOV and co-express mCherry for ground-truth identification, and mixed with another cell population expressing only EGFP. Cells are classified based on MFE response, demonstrating the possibility of identifying MagLOV reporters mixed with other, non-magnetic responsive

fluorescent reporters in the same spectral range, and under conditions where MagLOV produces only a small signal. **e**, Cropped view of single trenches, with cells circled in red as identified by a cell-segmentation algorithm. Scale bar, 2 μm. **f**, Time series of the 450-nm illumination fluorescence for the individual cells depicted in **e** over time, as a magnetic field $B_0 = 10$ mT is switched on and off (red line). Traces for each cell in the trench (grey) and the average over all the cells in the trench (black) are shown. **g**, Confusion scatter plots for classifying whether cells express MagLOV or EGFP based on magnetic response, taking presence of mCherry fluorescence (red highlights) as ground-truth control and using the MFE lock-in value as the predictor. The balanced accuracy by cells and by trenches is 0.99. **h**, The standard deviation $\sigma$ of the lock-in values in **g** is calculated between cells in each of the trenches (blue histogram), and between all cells in all trenches.

The formation of FMNH˙ from the excited triplet state, ³FMN*, probably proceeds via initial formation of FMN˙⁻ in an SCRP on a nanosecond timescale as in AsLOV $C_{450}A$ (ref. 33), followed by slow protonation. Alternatively, FMNH˙ could be formed by proton-coupled electron transfer. A proposed photoscheme is given in Fig. 3e, and a model based on this photoscheme is simulated in Supplementary Note 5, which successfully fits both the experimental MFE and ODMR data. In cryptochrome, the SCRP is formed by a cascade of electron transfers along a tryptophan tetrad[30]; however, a single donor aromatic amino acid can be sufficient, as demonstrated by the magnetosensitivity of cryptochrome-mimicking flavomaquettes[43,44] and FMN bound inside the bovine serum albumin protein[45]. Regarding the donor species, single-point mutations in AsLOV2 $C_{450}A$ leading to quenching of the emissive NMR signal suggest $W_{491}$ as the electron donor[46], which was corroborated by isotopic labelling of Trp residues[47]. However, given the extent of mutations in the variants studied here, we cannot confirm that $W_{491}$ is still the counter-radical. For example, in the structurally related iLOV-$Q_{489}$, derived from the *Arabidopsis thaliana* phototropin-2 (AtPhot2) LOV2 domain, transient absorption spectra revealed that a neutral tryptophan radical, Trp˙, is formed in conjunction with FMNH˙, and photoinduced flavin reduction in single-point mutations of selected tyrosine and tryptophan residues suggested that several amino acids might be involved in SCRP formation[48].

## Multiplexing and lock-in measurements

Our library of MFP variants shows differences in the rate and magnitude of response across both MFE and ODMR characterization (Fig. 2). Where such differences can be engineered orthogonally between variants, they open the possibility of using libraries of MFP reporters as a multiplexing tool to extract several signals from a single measurement modality (Fig. 4). As a demonstration, we characterized two cell populations expressing different MFP variants, for which distributions of MFE saturation timescales fit to single cells show strong separation (Fig. 4b). This enabled population decomposition when we applied a classification algorithm to a mixed population of about 2,000 cells (Fig. 4c). The classifier was trained on MFE traces normalized by amplitude (on a per-cell basis), meaning that it utilizes only the relative shape of the curves and not the magnitude of the MFE. Therefore classification is robust to scaling or offsets in the absolute brightness of the signal, as might be caused by scattering or autofluorescence (which pose practical limits on many sensing applications as described in Supplementary Note 6). Future application of this subpopulation labelling technique would benefit from engineering of variants with greater variation in dynamics such that the separation of histograms (as in Fig. 4b,c) is significantly greater than each population's intracellular variability.

Using a microfluidic set-up (in which populations of five to eight clonal cells are confined in individual trenches; Supplementary Note 7),

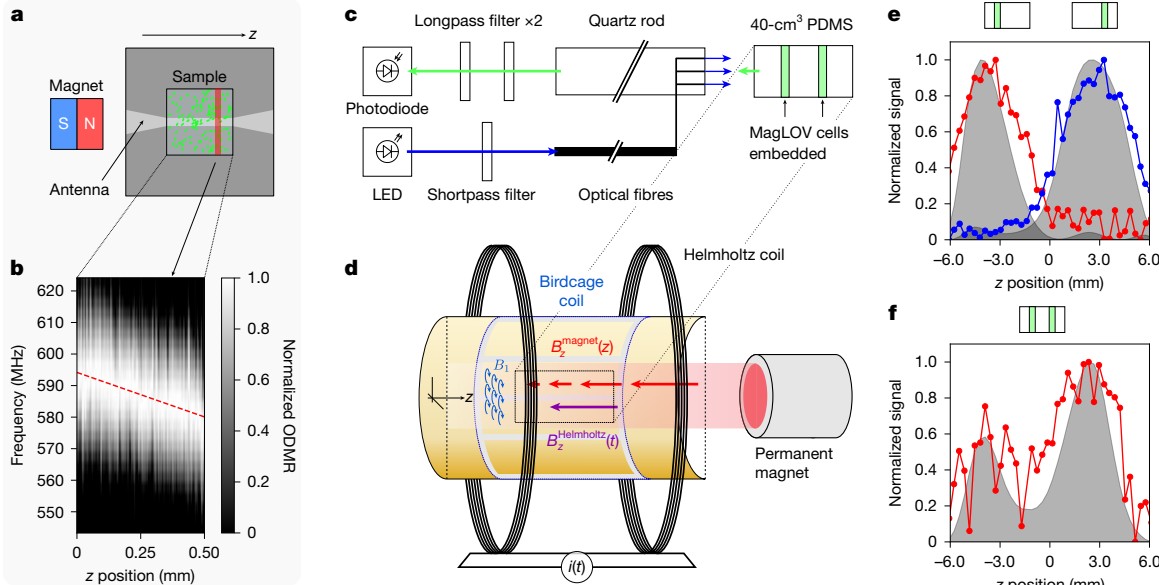

**Fig. 5 | Spatial localization using ODMR. a**, Schematic of the wide-field microscopy set-up for demonstrating localization of cells in a two-dimensional plane. The permanent magnet creates an approximately linear gradient field $B_0$ along the $z$ axis, perpendicular to the radio-frequency $B_1$ field (which rotates at the Larmor frequency around $z$) generated by the radio-frequency coil. The frequency of $B_1$ is scanned while the entire field of view is imaged. **b**, Subsequently, the images are divided into regions (red highlight in **a**) and integrated over that region. The integrated brightness versus frequency forms a one-dimensional spatial map—in this case, the sample is present in the entire field of view, thus we see a diagonal line, shifting by about 14 MHz over the 0.5-mm field of view as anticipated for the 1 mT mm$^{-1}$ field gradient. **c,d**, Schematic of the custom-built MRI set-up illustrating the optical illumination and collection paths (**c**) and the magnetic-field gradient inside the resonant MRI coil (**d**). **e,f**, We embedded cells expressing MagLOV 2 fast (chosen for its ODMR contrast, fast saturation and low overall brightness to make detection challenging) into a polydimethylsiloxane (PDMS) sample at two different positions along the coil axis separately (**e**) and simultaneously (**f**). Lock-in ODMR detection was used to locate the samples along the coil axis. The red and blue curves are raw measurements, and the grey shaded regions are after processing with a deconvolution algorithm (which uses the known ESR linewidth from Fig. 1 but makes no assumption about the number or location of peaks). The location of the samples is identifiable on their own, and resolvable via deconvolution together. Against a ground-truth separation of 7.5 mm, deconvolved individual samples had a calculated peak separation of 6.6 mm and the combined sample 6.1 mm. Using the individual sample data to calibrate the measurement yields a combined-sample distance estimate of approximately 6.9 mm.

we further investigated the sources of intracellular variability of MFP variants, and demonstrate the possibility of lock-in detection in weak signal environments. With MFE-based lock-in detection[45], cells with minimal MagLOV expression could be identified distinctly from cells expressing enhanced green fluorescent protein (EGFP)[49] with a balanced accuracy of about 0.99 (using a second fluorescence reporter, mCherry, as ground truth), with accuracy improving when averaging over a trench compared with distinguishing single cells (Fig. 4e–g). This approach allows variability to be attributed to inter-clonal or intra-clonal sources; we observe that mean intra-trench variability (quantified by standard deviation over cells in one trench) is approximately half that of the variability over all MagLOV-positive cells (Fig. 4h). This suggests that approximately one-quarter of the total variance arises from intra-clonal noise sources (for example, phenotypic variability over two or three generations, camera and accompanying measurement noise), with inter-clonal sources (longer-term phenotypic variability, variation in local environment) contributing the remaining three-quarters.

## Application to spatial localization

Methods for the spatial localization of fluorescence signals in biological samples such as cell cultures and tissue samples are of significant interest for both diagnostics and treatment development[50,51]. However, techniques based on localization using fluorescence, for example, fluorescence-modulated tomography, are challenging owing to the inherent scattering and absorbing nature of tissue, and the requirement to localize the fluorescence via detailed modelling and inversion of the optical signal[52]. As such, we sought to explore whether MagLOV could be used as a fluorescent marker localized by optically detected MRI.

First, we tested localizing MagLOV in the wide-field microscope set-up, using a permanent magnet to vary the resonance condition across the field of view (Fig. 5a). The sequence of images acquired during a radio-frequency $B_1$ frequency sweep was integrated over cross-sections in which the $B_0$ field is approximately constant, yielding an image (Fig. 5b) where the frequency of peak response ($y$ axis) denotes the position in space along the $z$ axis, which varies across a 0.5-mm field of view as anticipated for a field gradient of 1 mT mm$^{-1}$.

Next, we converted a preclinical MRI 28-mm-diameter 'birdcage' radio-frequency coil, used for creating a spatially highly homogeneous $B_1$ radio-frequency field at 500 MHz, into an optically detected fluorescence MRI instrument via integration of a fibre-coupled illumination system and imaging using a photodiode (Fig. 5c,d and Supplementary Note 8). It is noted that the photodiode is in effect a 'single-pixel' detector, meaning that it collects no spatial information and directional scattering of light would cause no reduction in information of final signal (apart from a possible decrease in absolute brightness). As the static $B_0$ field is swept, the radio-frequency field $B_1$ is switched on and off such that the ODMR contrast can be measured via lock-in detection. We found that good localization could be achieved when isolating single samples at different positions (Fig. 5e), and despite an increase in noise, deconvolution of the signal enabled two samples at different depths to be simultaneously localized (Fig. 5f) to within about 0.6 mm of a calibrated ground truth. We note that the generation of homogeneous radio-frequency fields at about 500 MHz within living systems (including humans) is an active area of research with known working designs for different species and anatomies[53]. In this way, our approach forms an alternative method for spatially resolved and scattering-insensitive sensing of genetically encodable fluorescent proteins.

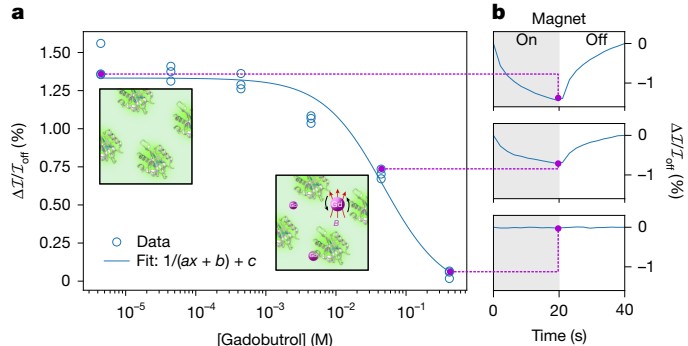

**Fig. 6 | Local spin environment sensing using MFE. a**, The MFE contrast of purified MagLOV 2 fast is measured at varying concentrations of the paramagnetic contrast agent gadobutrol, demonstrating a reciprocal dependence on concentration, consistent with spin relaxation and indicating that MagLOV is sensitive to its surrounding spin environment. For each gadobutrol concentration, the contrast was measured at three spatially separated fields of view. Insets are cartoon depictions of the solution with Gd (left) and without (right). Red arrows illustrate the local magnetic field generated by Gd; black arrows indicate rotational motion. **b**, The MFE time trace is shown for three points on the dose dependence curve of **a**. In general, we observed significantly lower MFEs for purified protein compared with in-cell measurements (see Supplementary Note 1 for further data).

## Application to microenvironment sensing

We next sought to determine whether MagLOV could function as a quantum sensor of its local environment. Although FMN is bound within the protein scaffold, we hypothesized that it might exhibit magneto-optical properties that depend on nearby paramagnetic species, analogous to nitrogen-vacancy centres in nanodiamond-based sensing approaches[2,54,55]. To test this, we diluted purified MagLOV 2 fast samples to 3.7 μM in solutions containing the paramagnetic contrast agent gadolinium (total spin $S = 7/2$; $Gd^{3+}$ chelated as the MRI contrast agent gadobutrol). We anticipated that these freely diffusing paramagnetic species would act like diffusing point dipoles, modulate the dipolar interactions of the radical pair, and lead to enhanced relaxation and characteristic attenuation of the MFE contrast. This hypothesis was confirmed by our measurements (Fig. 6), which show that, despite the constrained cofactor geometry, MagLOV exhibits a clear dose-dependent reduction in MFE signal with increasing paramagnetic ion impurity concentration. We note that there was no visual change in the microscopy images, nor any gadobutrol concentration-dependent difference in absolute brightness of the sample, indicating that the chelated gadolinium did not cause protein denaturation or aggregation, or FMN dissociation. This result implies that MFPs could be used as an in situ, in vivo quantum sensor, which, analogous to other quantum-sensing modalities, opens up possible applications as a sensor of the cellular microenvironment.

## Discussion

Directed evolution, enabled by straightforward fluorescence screening, has proved to be a powerful technique to engineer proteins exhibiting magneto-sensitive responses. The advent of stable, highly responsive magneto-sensitive proteins represents a paradigm shift; transitioning from quantum biological systems studied primarily for scientific interest towards engineerable tools with potential for widespread application. Previously, existing natural and engineered proteins (typically designed as model representatives of the cryptochrome) showed comparatively small responses to magnetic fields, required sophisticated experimental apparatus for study, did not exhibit measurable MFEs in living cells, were prone to rapid light-induced degradation, and were therefore unsuitable for biotechnological applications or

high-throughput set-ups required for directed evolution[22,27,43]. These challenges are simultaneously overcome by MagLOV. Furthermore, compared with other candidates for quantum biological sensing, two unique advantages of a protein-based system are: (1) that it is configurable, that is, significant engineering improvements can be made (relatively simply) by changing the DNA encoding the protein; and (2) that it can be endogenously expressed, enabling coupling of its expression to diverse genetic or chemical signals. MFPs therefore are the best of both worlds—enabling sensitive quantum measurements while also being highly amenable to engineering and cellular integration.

As we demonstrate, this system unlocks a host of measurement applications. First, modulating MagLOV fluorescence by applying a time-varying magnetic field enables multiplexing to expand the number of fluorescent reporters that can be distinguished in a single experiment. Meanwhile, lock-in could enable fluorescent protein measurements to be performed where previously not possible owing to poor signal quality, for example, if only small quantities of a fluorescent marker can be produced, or in tissue measurements where autofluorescence and scattering are limiting factors[15,45,56,57]. Second, we demonstrate that using ODMR it is possible to spatially localize fluorescence signals in a three-dimensional volume, utilizing the fact that resonance occurs only when the required conditions are met by (orthogonally controllable and tissue-penetrating) radio-frequency and magnetic fields. Finally, magnetic resonance sensing can be used to determine the presence of molecular species creating local magnetic noise in our protein's environment. This could be used to measure the presence of free radicals or paramagnetic metalloproteins, both critical to a number of physiological processes[55].

Although the properties of the MFPs we generated are superior to previously studied proteins that exhibit MFEs, their optimization is by no means complete. Much like fluorescent proteins, we expect that MFPs may be engineered to make general improvements, such as to solubility, photostability, spectral response and quantum yield, as well as further enhancing their MFE and ODMR properties. There also remains significant opportunity for mechanistic investigation utilizing the wide array of techniques previously applied to biological systems exhibiting MFEs and ESR[22,31,34]. Crucially, mechanistic understanding and high-throughput bioengineering (similar to and more advanced than we demonstrate here) can go hand in hand—for instance, enabling the creation of rational design tools that can optimize MFE and ODMR properties for a specific application. Finally, we hope that the development of MFPs can serve as the starting point for a class of magnetically controlled biological actuators, whereby application of a local magnetic field can be coupled to downstream cellular effects—such a technology would be of significant biomedical and biotechnological utility.

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

## Methods

### Strains

T7 Express *Escherichia coli* (New England Biolabs, C2566) were used for MFE and ODMR experiments in Figs. 1–3. *E. coli* MG1655 cells were used for multiplexing and mother machine lock-in experiments in Fig. 4, owing to its effectiveness as a heterologous gene expression strain, and past optimization of mother machine chip loading protocols for this host. MG1655 was also used for Fig. 5. NEB Dh5α *E. coli* was used for plasmid production and genetic engineering steps, but not experimental data collection.

### Cell media

TB auto-induction medium (Formedium, AIMTB0260) was used for growth of liquid cultures of T7 Express strains. LB medium (Formedium, LBX0102) was used for growth of MG1655 strains in liquid cultures and agar plates (Formedium, AGR10). M9 medium (Formedium, MMS0102; supplemented with 0.2 g l$^{-1}$ Pluronic F-127 (Sigma Aldrich, P2443-250G), 0.34 g l$^{-1}$ thiamine hydrochloride, 0.4% glucose, 0.2% casamino acids, 2 mM MgSO$_4$, 0.1 mM CaCl$_2$, 50 mg l$^{-1}$ EDTA Na$_2$.2H$_2$O, 5 mg l$^{-1}$ FeCl$_3$, 0.84 mg l$^{-1}$ ZnCl$_2$, 0.13 mg l$^{-1}$ CuCl$_2$.2H$_2$O, 0.05 mg l$^{-1}$ CoCl$_2$, 0.1 mg l$^{-1}$ H$_3$BO$_3$ and 0.016 mg l$^{-1}$ MnCl$_2$.4H$_2$O) was used for mother machine experiments. Antibiotic stocks were prepared as follows: 100 mg ml$^{-1}$ carbenicillin (Formedium, CAR0025) dissolved in 50% (v/v) ethanol and 20 mg ml$^{-1}$ chloramphenicol (Fisher, 10368030) dissolved in 100% (v/v) ethanol; stocks were diluted 1,000× for experiments.

### Plasmid construction

Whole plasmid sequencing was performed by Plasmidsaurus using Oxford Nanopore Technology.

Derivative plasmids of those generated by directed evolution (for example, MagLOV + mCherry) were constructed using the EcoFlex kit (Addgene kit 1000000080) following standard EcoFlex protocols[61]. Level 1 assemblies were performed using NEB BsaI-HFv2, NEB T4 DNA Ligase Reaction Buffer (B0202S) and Thermo Scientific T4 DNA Ligase (EL0011). Level 2 assemblies were performed using NEBridge Golden Gate Assembly Kit BsmBI-v2 and NEB T4 DNA Ligase Reaction Buffer (B0202S).

Level 1 PCR cycling protocol: 30× (5 min at 37 °C → 5 min at 16 °C) → 10 min at 60 °C → 20 min at 80 °C.

Level 2 overnight PCR cycling protocol: 65× (5 min at 42 °C → 5 min at 16 °C) → 10 min at 60 °C → 20 min at 80 °C.

pTU1 was used as a negative control plasmid (Supplementary Note 2); it is part of the EcoFlex kit.

pRSET-AsLOV_R2, pRSET-MagLOV, pRSET-MagLOV-2 and pRSET-MagLOV-2_R11-f plasmids (used for Figs. 1–3) contain MFP variants created via directed evolution (described below) under control of T7 promoters. It is noted that despite the T7 promoter being chemically inducible in its host strain, experiments were performed without induction (that is, transcription was owing to leaky expression).

pVS-01-03_EGFP expressing EGFP was used for microfluidic experiments. It was assembled solely from EcoFlex parts (pTU1-B-RFP, pBP-J23108, pBP-pET-RBS, pBP-ORF-eGFP and pBP-BBa-B0012).

pVS-02-04_mCherry+MagLOV was used for microfluidic experiments. EcoFlex Golden Gate assembly was used to create a plasmid that expresses both MagLOV and mCherry. In a level 1 reaction, pVS-01-01_mCherry and pVS-01-02_MagLOV were assembled using the following bioparts from the EcoFlex kit. pVS-01-01_mCherry: pTU1-A-lacZ, pBP-J23108, pBP-pET-RBS, pBP-ORF-mCherry, pBP-BBa_B0012. pVS-01-02_MagLOV: pTU1-B-RFP, pBP-J23119, pBP-pET-RBS, BP-01-MagLOV, pBP-BBa_B0012. The BP-01_MagLOV sequence with overlaps was gained via PCR using NEB Q5 polymerase with pGA-01-01 as template using primers MagLOV_EcoFlex_FWD and MagLOV_EcoFlex_REV. Both plasmids were combined in a subsequent Level 2 EcoFlex reaction

with the pTU2-a backbone, leading to pVS-02-04_mCherry+MagLOV. The predicted translation rate is 250.

pRH-01-17, pGA-01-47 and pGA-01-49 were used for multiplexing experiments and spatial localization. The coding sequences (CDSs) for AsLOV2 R5, MagLOV 2 and MagLOV 2 fast, respectively, were codon optimized and the ribosome binding site (RBS) designed to maximize constitutive expression using the CDS Calculator and RBS Calculator tools of ref. 62. The sequence was synthesized as a gene fragment (Twist Bioscience) and assembled into a level 1 plasmid as above, again using the J23119 promoter.

MFP expression levels were estimated for constructs using ref. 62. For pVS-02-04, an unoptimized codon sequence and RBS were used, resulting in a predicted translation rate in its expression strain (*E. coli* MG1655) of 261. For pRSET constructs, the predicted translation rate in expression strains (*E. coli* NEB T7) is $50 \times 10^3$. For pRH-01-17, pGA-01-47 and pGA-01-49, the optimized codon sequence and RBS yielded a predicted translation rate in *E. coli* MG1655 of $10^6$; as such this represents a 20× increase in predicted rate over the pRSET strains and a 4,000× increase in predicted translation over pVS-02-04. It is noted that these predictions also do not account for the dual expression in pVS-02-04 of mCherry.

### Directed evolution of MFPs

We previously reported the directed evolution of MFPs resulting in MagLOV[8]. Directed evolution of MFP variants was initiated with AsLOV2 C450A[39], a protein originally isolated from the common oat *Avena sativa*. The mutagenesis protocol was based on methods developed previously[63]. In particular, we used 142 PCR reactions with NNK semi-random primer pairs (supplied by Eton Bioscience) to make all single amino acid mutants of AsLOV2 (404–546) C450A at all locations, which produces a library containing 2,982 protein variants, although (owing to the pooled nature of the screen) it is likely that not all variants are present for each round. This library was transformed into *E. coli* strain BL21(DE3) and spread across about ten LB-agar plates for screening. After transformation, plates were left for 2 days at room temperature (25 °C) as this was observed to lead to stronger magnetoresponses. Screening used the same fluorescence photography system described in ref. 63, with addition of an electromagnet (KK-P80/10, Kaka Electric) below the sample, which was turned on/off every 15 seconds using an Arduino Uno while acquiring successive images. After each screen, images were processed to identify the single colony (across all plates) with the largest fractional change in fluorescence between the magnet on/off conditions. The selected colony was picked (manually) from the corresponding plate, and used as the basis for the next round of mutagenesis and selection. Subsequent mutagenesis rounds used the same primers to avoid re-making the primer library with each mutation (which we hypothesize selects against multiple consecutive nearby mutations), except before rounds leading to AsLOV R5, MagLOV R7 and MagLOV 2, at which point primers in the library that overlapped mutated sites were updated to the current variant's sequence. In total, 11 rounds of mutagenesis were undertaken, all selecting for amplitude of MFE with the exception of round 11 (MagLOV 2 fast) where mutants from round 10 were selected based on fast response time (quantified as the mutant with the largest contrast measured in the first 100 ms following magnetic-field change). After each round of mutagenesis, selected variants were sequenced using the Sanger method (supplied by Quintara Biosciences). The variants measured in this paper and some intermediaries are provided in Supplementary Note 3.

### Wide-field MFE and ODMR characterization

Plasmids expressing EGFP, MFP variants or negative control plasmids expressing only antibiotic selection markers were used to transform *E. coli* NEB T7 via heatshock at 42 °C for 45 seconds. Colonies were grown overnight at 37 °C on Agar plates, then a single colony was

picked, resuspended in 1 ml LB media, spread onto a plate, and grown at 37 °C for 24 hours followed by 24 hours at room temperature to form a lawn of cells. Cells from this lawn were then resuspended in PBS buffer.

Both MFE and ODMR imaging were performed using a custom-built wide-field epifluorescence microscope (Supplementary Notes 10 and 11). From cells suspended in PBS buffer, an approximately 1-µl droplet was confined between two glass coverslips (SLS MIC2162) atop a stripline antenna printed circuit board (Supplementary Note 12). The antenna printed circuit board was placed inverted on the microscope stage and either an electromagnet (for MFE) or a permanent magnet (for ODMR) was positioned above. The antenna printed circuit board was used to mount the MFE experiments (despite not delivering any $B_1$ field) to that ensure lighting conditions were consistent with those in ODMR experiments. For MFE experiments, the electromagnet supplied a static field of 0 mT or 10 mT at the sample. For ODMR experiments, the permanent magnet supplied a static field ($B_0 \approx 20$ mT) in the $z$ direction (Supplementary Note 13), perpendicular to the radio-frequency field ($B_1 \approx 0.2$ mT) supplied by the stripline antenna (Supplementary Note 14). It is noted that the imaging step and exposure times (Supplementary Note 11) were maintained across experiments unless otherwise stated, whereas light-emitting-diode power levels (Supplementary Notes 9 and 15) were varied between samples to account for differing expression levels of MFPs.

Data processing was performed using Python (v3.11.11), SciPy (v1.15.1)[64], NumPy (v.126.4)[65], scikit-learn (v1.6.1)[66] and scikit-image (v0.20.0)[66].

## Calculation of MFE and ODMR effect sizes

We define the MFE by $\Delta\mathcal{I}/\mathcal{I}_{off}$, where $\mathcal{I}$ is the fluorescence intensity, $\mathcal{I}_{off}$ is the fluorescence intensity immediately before switching the magnet on, and $\Delta\mathcal{I} = \mathcal{I} - \mathcal{I}_{off}$. Similarly, for ODMR measurements, the detrended signal is defined to be $\Delta\mathcal{I}/\mathcal{I}_{bg}$, where $\mathcal{I}_{bg}$ is a background curve fit to the data as described in Supplementary Note 9, and $\Delta\mathcal{I} = \mathcal{I} - \mathcal{I}_{bg}$.

## Spectral measurements

Wavelength-resolved MFE measurements (Fig. 3 and Supplementary Note 4) were performed in bulk using a home-built cuvette-based fluorescence spectrometer described in ref. 24. In brief, a 450-nm laser diode was used to excite the sample, which was housed inside custom-built Helmholtz coils. The emission was collected through a lens pair and dispersed by a spectrograph (Andor Holospec) onto a charge-coupled-device array (Andor iDus420).

Cells were resuspended in PBS buffer at an optical density of about 0.3 and placed in a quartz fluorescence cuvette (Hellma). Similarly to the wide-field measurements, a magnetic field was switched on and off with field strength 10 mT and a period of 20 seconds (10 seconds on, 10 seconds off). The sample was illuminated at 450 nm (Oxxius LBX-450), at roughly 1 kW m$^{-2}$, and the emission was filtered using a 458-nm longpass filter (RazorEdge ultrasteep).

Excitation spectra in Fig. 3c and Supplementary Note 4 were recorded using an Edinburgh Instruments FS5 Spectrofluorometer, using a xenon lamp as an excitation source. Emission spectra (Fig. 3a) and time-resolved emission spectra (Supplementary Note 4) were recorded on the same instrument, but using an HPL450 (450 nm) for excitation. It is noted that this instrument set-up suffers from laser excitation artefacts, which create the small peaks near 550 nm and 650 nm.

## Multiplexing using MFE

To demonstrate the potential of using MFE for multiplexing in fluorescence microscopy, we first cloned strains of AsLOV R5 (one of the variants evolved between AsLOV R2 and MagLOV; Supplementary Note 3) and MagLOV 2 into *E. coli* MG1655. Cells were grown

in liquid culture from −80 °C freezer stocks, then resuspended in PBS and made up to equal concentration as measured by optical density (OD600). Individually, and as a 1:1 mixture, monolayers of cells were sandwiched between a glass slide and cover for MFE imaging.

We initially characterized each variant in isolation, measuring fluorescence traces for approximately 2,000 cells in a field of view (Fig. 4a). The same processing was performed as in Fig. 2 but to individual cells, yielding a value of $\tau$ (the MFE saturation timescale) for each cell, which allows measurement of each sample's intrapopulation variability (Fig. 4b).

To perform the population decomposition, we trained a machine-learning classifier (XGBoost[67]) on the dynamic data (that is, fluorescence versus time) used to generate Fig. 4b. Before training, and for classification, the MFE response ($\Delta\mathcal{I}/\mathcal{I}_{off}$) curves of each cell were normalized to range from 0 to 1. Without further training, this classifier was used to classify a combined population of cells (Fig. 4b) mixed with 1:1 ratio (determined by OD600). The ratio of the two variants by classification was R5:ML2 = 0.9:1, which probably differs from the anticipated 1:1 ratio owing to both classifier accuracy and typical measurement errors expected from the use of OD600 to quantify cell counts[68].

## Lock-in detection using MFE

Cells expressing EGFP[49] and cells co-expressing mCherry[69] with very weak MagLOV expression (expression level predicted[62] at about 0.025% of that in multiplexing experiments) were grown in liquid culture, then mixed and loaded into a microfluidic 'mother machine' chip consisting of two rows of evenly spaced trenches fed by a central channel bringing fresh media into the system and carrying excess cells out. After a few hours of growth, each trench was filled only with cells whose ancestor is the cell at the closed end of the trench (that is, they may be considered clonal; Fig. 4d). The cells were imaged using the same wide-field fluorescence microscopy set-up as described previously. Microfluidic manufacturing and set-up techniques are described in Supplementary Note 7. Algorithms for cell segmentation and image processing and quantification are described in Supplementary Note 16. Quantitative lock-in classification results are provided Supplementary Note 17. In Fig. 4g, the classifications are shown including the decision boundaries based on the control (565-nm fluorescence) and based on the lock-in value. The graphs can be interpreted as confusion matrices: top-left quadrant are false negatives, top-right quadrant are true positives, bottom-left quadrant are true negatives, and bottom-right quadrant are false positives.

## Spatial localization using ODMR

The fluorescence MRI instrument (described in detail in Supplementary Note 8) used a permanent neodymium rare-earth magnet to impose a static magnetic field, $B_0$, which varied linearly along the $z$ axis with a gradient strength of approximately 0.95 T m$^{-1}$ at the radio-frequency isocentre of the coil used. Our methodology assumes that (1) this gradient is uniform within the coil and (2) there is no variation in $x$–$y$ planes; violation of these assumptions would degrade the localization accuracy. The imaging isocentre is both spatially the centre of the radio-frequency coil's sensitive area and was chosen to be at 17.8 mT corresponding to an ESR Larmor frequency of $f \approx 500$ MHz, the coil's resonant frequency. The magnetic field was modulated axially by a Helmholtz coil such that $B_z = B_z^{magnet}(z) + B_z^{Helmholtz}(t)$ where $t$ is time. By varying the Helmholtz coil current between ±1 A, we were able to shift the effective magnetic field by ±5.87 mT around the resonant condition, providing a total field of view of approximately 12 mT along the gradient direction (see Supplementary Note 8 for a detailed calculation). The device is therefore able to scan spatially in one dimension as the Larmor frequency remains fixed at $f = \bar{\gamma}_e B_z(z, t)$, where

different spatial positions are brought into resonance through current modulation while maintaining fixed radio-frequency excitation.

To simulate a three-dimensional volumetric sample, we cast a 40-cm$^3$ volume of PDMS (matching the region of uniformity of the radio-frequency coil) with empty cylinders of 0.4-mm-diameter embedded in it perpendicular to the $B_0$ gradient and separated in $z$ by 7.5 mm. We filled these cylinders with cells expressing MagLOV 2 fast (centrifuge-concentrated liquid culture), measuring at different depths both individually and together. Intensity data were processed (grey solid area in Fig. 5e,f) using a Lucy–Richardson deconvolution[64] with a Gaussian impulse response function of fixed width 3 mm, which is estimated based on the ODMR linewidth of 80 MHz (see Supplementary Note 8 for further details). It is noted that although the deconvolution algorithm uses the known ODMR linewidth (and hence point-spread function), it does not make any assumption about the number or location of peaks.

This fluorescence MRI instrument was designed and built to achieve proof of concept and is limited in terms of signal-to-noise performance of the optical path and photodiode sensor, as well as $B_0$ field uniformity in the target volume. Future development could significantly improve its capabilities via implementing full three-dimensional control of $B_0$ field gradients, an endoscope-type imaging system to collect light with $x$–$y$ spatial information from the target volume without sensor perturbation by radio frequency or magnetic field, addition of fast optical lock-in to the fluorescence excitation or measurement, optical path re-design to improve collection and filtering efficiency and reduce stray light, and indeed directed evolution of variants with faster ODMR response dynamics to allow averaging over more on/off cycles.

## Microenvironment sensing

Purified MagLOV 2 fast solutions for the data in Fig. 6 were prepared as follows. It is noted that all MagLOV proteins expressed include a His-tag at the N-terminus. Protein was purified following using HisPur Cobalt Resin (Thermo Scientific) following the manufacturer's protocol. Purified protein was suspended in PBS buffer with 30% glycerol and stored at −80 °C.

We estimated the concentration the purified protein samples via the method used in ref. 70 to determine the extinction coefficient of LOV-based fluorescent proteins. First, purified protein was heated to 95 °C for 5 minutes to dissociate the FMN from the protein (we assume that one FMN corresponds to one protein, as free FMN should be removed by the protein purification process). The concentration of flavin (and hence protein) was then determined by performing a serial dilution and measuring absorbance at 450 nanometres ($A_{450nm}$), using the free FMN extinction coefficient $\epsilon_{FMN} = 12{,}200$ M$^{-1}$ cm$^{-1}$ (ref. 71).

Contrast agent experiments were performed on the wide-field microscope configured for MFE detection, using a six-chamber microfluidic chip (ibdi μ-Slide VI 0.5 Glass Bottom) to hold samples of gadobutrol (CRS Y0001803) diluted in PBS buffer in serial dilution (MagLOV concentration the same for all conditions). Measurements were acquired in a randomized sequence (to compensate for any stray light photobleaching or time and sequence correlated effects) by programming 6 × 3 fields of view (6 chambers of differing concentrations, 3 fields of view in each chamber), then automatically performing an MFE measurement acquisition at each field of view. For each MFE acquisition, 10 periods of duration 40 seconds (20 seconds magnet on, 20 seconds off) were acquired at 450-nm light-emitting-diode intensity 280 mW cm$^{-2}$, 100-ms image exposure time and 10-mT magnetic-field strength. Optimization of experimental protocols for spin-relaxometry-based sensing using MagLOV or other radical-pair fluorescent proteins will be required for broader application; here we demonstrate simply that the spin-radical pair of MagLOV is indeed sensitive to its surroundings despite the flavin being bound.

To consider the expected effect of paramagnetic species upon the MagLOV MFE more quantitatively, we note that paramagnetic impurities can be modelled as stochastic point dipoles that modify the $T_1$ or $T_2$ relaxation rates in radical-pair kinetics[72], thereby changing the contrast. We therefore anticipated that a characteristic timescale for this process, $\tau$, would scale as $\tau \approx \frac{1}{k_{STD}} + \left(\frac{1}{T} + R[\text{Gd}^{3+}]\right)^{-1}$ where $k_{STD}$ is a stochastic decoherence rate, $T$ is a semiempirical $T_1$ or $T_2$ relaxation time, and $R$ the effective relaxivity of the paramagnetic impurity. We therefore expect contrast to fit a functional form of approximately $(a[x] + b)^{-1} + c$, and indeed this is consistent with Fig. 6.

## Reporting summary

Further information on research design is available in the Nature Portfolio Reporting Summary linked to this article.

## Data availability

The experimental data that support the findings of this study are available at https://doi.org/10.25446/oxford.30344995. Plasmid sequencing data have been made available on the European Nucleotide Archive (ENA) with the identifier PRJEB83586.

## Code availability

Code required to generate the figures are available at https://doi.org/10.25446/oxford.30344995.

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

**Acknowledgements** We thank K. Henbest, E. Vatai and L. Gerhards for discussions; D. Cubbin for help with the ultraviolet–visible characterization; I. Robertson and P. Reineck for assistance with proof-of-concept experiments; C. Carr and G. Mazur for lending of radio-frequency equipment; P. Freemont for gift of an EcoFlex kit; and the ChimeraX[73] team for protein-structure rendering tools used in Fig. 1. G.A. and S.S. were supported by funding from the Biotechnology and Biological Sciences Research Council (UKRI-BBSRC; grant number BB/T008784/1). J.J. and V.T.-F. were supported by funding from the Engineering and Physical Sciences Research Council (UKRI-EPSRC; grant number EP/W524311/1). I.K. and H.S. are supported in part by the UKRI-EPSRC under the EEBio Programme Grant, EP/Y014073/1, and EP/X017982/1 and UKRI-BBSRC (grant number BB/W012642/1). R.H. is supported by funding from the UKRI-EPSRC (grant number EP/Y034791/1). A.Š., L.M.A. and C.R.T. are supported by the European Research Council under the European Union's Horizon 2020 research and innovation programme, grant agreement number 810002, Synergy Grant: 'QuantumBirds'. C.R.T. thanks the US Army. J.J.M. acknowledges support from the Novo Fonden (NNF21OC0068683). H.S. recognizes support from the Philip Leverhulme Prize.

**Author contributions** G.A., H.S., A.Y. and M.I. conceived of the study. G.A., A.Š., A.Y., M.I., C.R.T., J.M.M., J.-P.T. and H.S. designed the experiments. G.A., V.S., R.H., I.K., J.J., K.S., S.S. and V.T.-F. prepared samples and strains, and performed the microscopy experiments and analyses. M.I. performed the directed-evolution experiments. A.Š. performed the spectroscopy experiments.

G.A., J.M.M. and H.S. designed and built the MRI set-up. G.A., H.S., I.K., J.J. and K.S. designed and built the microscopy and microfluidic platforms and data processing tools. L.M.A., A.Š., G.A., J.-P.T. and C.R.T. developed the simulations and theory for the SCRP mechanism. G.A. and H.S. coordinated work and wrote the paper with input from all authors. H.S. supervised the project.

**Competing interests** M.I. is a cofounder and shareholder of Nonfiction Laboratories, a start-up company developing magnetogenetic control for therapeutic proteins.

**Additional information**
**Correspondence and requests for materials** should be addressed to Gabriel Abrahams or Harrison Steel.

# Reporting Summary

## Statistics

For all statistical analyses, confirm that the following items are present in the figure legend, table legend, main text, or Methods section.

| n/a | Confirmed | |
|---|---|---|
| ☐ | ☒ | The exact sample size (*n*) for each experimental group/condition, given as a discrete number and unit of measurement |
| ☐ | ☒ | A statement on whether measurements were taken from distinct samples or whether the same sample was measured repeatedly |
| ☒ | ☐ | The statistical test(s) used AND whether they are one- or two-sided<br>*Only common tests should be described solely by name; describe more complex techniques in the Methods section.* |
| ☒ | ☐ | A description of all covariates tested |
| ☒ | ☐ | A description of any assumptions or corrections, such as tests of normality and adjustment for multiple comparisons |
| ☐ | ☒ | A full description of the statistical parameters including central tendency (e.g. means) or other basic estimates (e.g. regression coefficient) AND variation (e.g. standard deviation) or associated estimates of uncertainty (e.g. confidence intervals) |
| ☒ | ☐ | For null hypothesis testing, the test statistic (e.g. *F*, *t*, *r*) with confidence intervals, effect sizes, degrees of freedom and *P* value noted<br>*Give P values as exact values whenever suitable.* |
| ☒ | ☐ | For Bayesian analysis, information on the choice of priors and Markov chain Monte Carlo settings |
| ☒ | ☐ | For hierarchical and complex designs, identification of the appropriate level for tests and full reporting of outcomes |
| ☒ | ☐ | Estimates of effect sizes (e.g. Cohen's *d*, Pearson's *r*), indicating how they were calculated |

*Our web collection on statistics for biologists contains articles on many of the points above.*

## Software and code

Policy information about availability of computer code

| Data collection | Data collection code used Python (v3.11.11), SciPy (v1.15.1) , NumPy (v.126.4) , scikit-learn (v1.6.1) and scikit-image (v0.20.0) in addition to manufacturer-provided drivers and software for instruments used in the work (instruments identified in methods/supplementary notes). |
|---|---|
| Data analysis | Data analysis code used the DeLTA library (v2), Python (v3.11.11), SciPy (v1.15.1) , NumPy (v.126.4) , scikit-learn (v1.6.1) and scikit-image (v0.20.0). Code required to generate figures has been uploaded alongside data as per data availability statement below. |

For manuscripts utilizing custom algorithms or software that are central to the research but not yet described in published literature, software must be made available to editors and reviewers. We strongly encourage code deposition in a community repository (e.g. GitHub). See the Nature Portfolio guidelines for submitting code & software for further information.

## Data

Policy information about availability of data

All manuscripts must include a data availability statement. This statement should provide the following information, where applicable:

- Accession codes, unique identifiers, or web links for publicly available datasets
- A description of any restrictions on data availability
- For clinical datasets or third party data, please ensure that the statement adheres to our policy

| The experimental data that support the findings of this study and code required to generate the figures are available at \href{https://doi.org/10.25446/ |
|---|

oxford.30344995}{10.5281/oxford.30344995}. Plasmid sequencing data have been made available on the European Nucleotide Archive (ENA) with the identifier \href{https://www.ebi.ac.uk/ena/browser/view/PRJEB83586}{PRJEB83586}

## Research involving human participants, their data, or biological material

Policy information about studies with human participants or human data. See also policy information about sex, gender (identity/presentation), and sexual orientation and race, ethnicity and racism.

| | |
|---|---|
| Reporting on sex and gender | NA |
| Reporting on race, ethnicity, or other socially relevant groupings | NA |
| Population characteristics | NA |
| Recruitment | NA |
| Ethics oversight | NA |

Note that full information on the approval of the study protocol must also be provided in the manuscript.

# Field-specific reporting

Please select the one below that is the best fit for your research. If you are not sure, read the appropriate sections before making your selection.

☒ Life sciences        ☐ Behavioural & social sciences        ☐ Ecological, evolutionary & environmental sciences

For a reference copy of the document with all sections, see nature.com/documents/nr-reporting-summary-flat.pdf

# Life sciences study design

All studies must disclose on these points even when the disclosure is negative.

| | |
|---|---|
| Sample size | For Figs 1, 2 and 4 sample size was determined by the number of cells in a field of view (~1000) that could be fit while clearly distinguishable. Fig 4 panels H-G was due to the number of cells that were randomly successfully loaded into the mother machine, with a field of view selected to maximise the number of visible cells. Fig 3 was a bulk spectroscopy measurement and therefore sample size of 1 was sufficient. Fig 6 was also bulk measurements, repeated in triplicate for the purpose of eliminating stray photobleaching as an effect, but not for any statistical purpose. For all figures experiments were repeated multiple times as resources and sample availability permitted, and error bars included (specified per-figure in the manuscript). Note that no statistical testing methodology which explicitly depends on numbers of samples was performed in the work. |
| Data exclusions | Data with the clearest signal-to-noise was chosen for presentation in the final manuscript, not based on pre-established criteria. |
| Replication | Experiments were repeated 2-3 times on different days starting from fresh growth of bacteria, as time and resources allowed. Experiments were replicated in the same laboratory using the same equipment. Only one iteration of each experiment was included (as above, chosen based on signal-to-noise). All results were successfully reproduced, however we note that microfluidic experiments are highly chip- and setup-dependent therefore reasonable reproduction requires precise adherence to protocol and set-up parameters. |
| Randomization | Not relevant. |
| Blinding | Not relevant. All analysis was performed algorithmically with the same code applied to each group in a given experiment. |

# Reporting for specific materials, systems and methods

We require information from authors about some types of materials, experimental systems and methods used in many studies. Here, indicate whether each material, system or method listed is relevant to your study. If you are not sure if a list item applies to your research, read the appropriate section before selecting a response.

## Materials & experimental systems

| n/a | Involved in the study |
|---|---|
| ☒ ☐ | Antibodies |
| ☒ ☐ | Eukaryotic cell lines |
| ☒ ☐ | Palaeontology and archaeology |
| ☒ ☐ | Animals and other organisms |
| ☒ ☐ | Clinical data |
| ☒ ☐ | Dual use research of concern |
| ☒ ☐ | Plants |

## Methods

| n/a | Involved in the study |
|---|---|
| ☒ ☐ | ChIP-seq |
| ☒ ☐ | Flow cytometry |
| ☒ ☐ | MRI-based neuroimaging |

## Plants

| | |
|---|---|
| Seed stocks | *Report on the source of all seed stocks or other plant material used. If applicable, state the seed stock centre and catalogue number. If plant specimens were collected from the field, describe the collection location, date and sampling procedures.* |
| Novel plant genotypes | *Describe the methods by which all novel plant genotypes were produced. This includes those generated by transgenic approaches, gene editing, chemical/radiation-based mutagenesis and hybridization. For transgenic lines, describe the transformation method, the number of independent lines analyzed and the generation upon which experiments were performed. For gene-edited lines, describe the editor used, the endogenous sequence targeted for editing, the targeting guide RNA sequence (if applicable) and how the editor was applied.* |
| Authentication | *Describe any authentication procedures for each seed stock used or novel genotype generated. Describe any experiments used to assess the effect of a mutation and, where applicable, how potential secondary effects (e.g. second site T-DNA insertions, mosiacism, off-target gene editing) were examined.* |

