## [Peer Review File · Nature]

Quantum Spin Resonance in Engineered Proteins for Multimodal Sensing

Corresponding Author: Professor Harrison Steel

Version 0:

Reviewer comments:

Referee #1

(Remarks to the Author)

The manuscript by Abrahams et al describes the physico-chemical mechanism underlying the response of flavin-associated fluorescent proteins to magnetic fields, demonstrates the use of directed evolution to alter this response, and illustrates potential applications in multiplexed imaging and lock-in amplification. Overall, this is an important and fascinating study that will no doubt garner interest from a broad audience. The overall story and data are convincing. I just have a few suggestions for the authors.

The claim that the R11f mutant has a faster saturation rate is not very evident based on the data shown in Fig. 4. It would be great to see some fitting or other quantification.

The authors observe that the ODMR response is much larger (as a %) in single cells than it is in the bulk. The authors attribute this to "increased background fluorescence, especially in the case of cell conglomerates". Can you be more specific? Autofluorescence of what? Why should it be different in a more concentrated solution of the same thing? Perhaps there are more complex (and interesting) phenomena at play, and this % change seems like a significant practical issue.

The authors speculate about potential applications to MRI-like spatial readout of fluorescence and actuation rather than imaging. I think it's fine to have such speculations. However, in my opinion, anything that is not actually demonstrated in the study belongs in the discussion section and not the abstract or introduction.

Speaking of speculation (which I admit is fun!), would an MRI-like readout be limited to 1 dimension or are there prospects for encoding additional dimensions via a pulse sequence analogous to what is done in MRI? Are there fundamental physical limitations on lifetimes, etc, that would make this difficult?

Also on speculation, it is not immediately clear to me how the mechanism described here could be used for protein actuation, since it requires the build-up of states through a fluorescence cycling process, if I understood it correctly.

Referee #2

(Remarks to the Author)

This manuscript by Abrahams et al. presents the development of magneto-sensitive fluorescent proteins based on spin-correlated radical pairs with flavin molecules. The engineered proteins exhibit enhanced magnetic field effects (MFE) in response to a static magnetic field (10–30 mT) and demonstrate improved optically detected magnetic resonance (ODMR) characteristics under an alternating magnetic field (~0.3 mT) at resonance frequencies corresponding to hyperspectral energy level splitting. Notably, the proteins achieve an intensity contrast as high as 30% in solution (Fig. 1D), 4% in a single bacterium (Fig. 2E), and 0.05% in bacterial culture. The manuscript also explores the magnetic sensitivity of several genetic variants and successfully demonstrates multiplexed detection of two distinct fluorescent proteins. Potential applications,

such as spatial localization, local magnetic noise sensing, and protein binding assays, are discussed.

Overall, the approach is technical sound. The observed enhancement of MFE and ODMR effects in live cells is particularly noteworthy. However, the manuscript lacks a quantitative comparison of sensitivity (contrast) relative to other organic molecules and inorganic nanocrystals, making it challenging to assess the level of innovation. From an application standpoint, the feasibility and sensitivity of the magnetic resonance-based approach, as well as its advantages over existing methods, remain unclear. The figures could be improved to enhance clarity and visual presentation.

Questions and Comments:

[Mechanisms]

- Fig. S6 could replace Fig. 1B and be cited in the second paragraph of Page 6.
- What are the typical values of key transition rates—fluorescence, intersystem crossing (ISC), back electron transfer rate (k_{BET}), and reaction rates (k_{A} , k_{D})—for the molecules studied in this work?
- How do the MRE and ODMR signal strengths ($\Delta\%$) vary as a function of the static magnetic field strength (B_0)?
- Providing theoretical equations for I2 and I3 (in Fig. S6) as functions of B_0 and B_1 would be beneficial. Additionally, a direct comparison between theoretical predictions and experimentally observed sensitivities would strengthen the analysis.

[Signal Contrast and Sensitivity]

- The manuscript reports an ODMR contrast of 4% from a single live cell in culture (Fig. 1E), where the number of proteins is estimated to be at most 10^5 per cell. This sensitivity is described as three orders of magnitude better than current electron spin resonance (ESR) techniques. Could the authors provide additional data and a statistical analysis across different single cells to confirm this improvement?
- The contrast of 4% in single cells decreases to 0.05% in cell conglomerates (Fig. 2B), attributed to increased background fluorescence. What is the autofluorescence intensity (e.g., photons per square micrometer) in multi-cell conglomerates compared to isolated single cells? Further investigation into the underlying reasons for this contrast reduction would be valuable. Are the cell conglomerates a monolayer or multi cell layers?
- Considering potential magnetic-field sensing, what would be the detection sensitivity values of static (B_0) and oscillating (B_1) magnetic fields, expressed, for example, in $\text{mT}/\sqrt{\text{Hz}}$?
- How do the measured contrast values compare with previous studies on ODMR and MRE using organic molecules and inorganic materials?
- What are the optical illumination intensity levels experimentally used for different samples?
- What are the signal integration times for the different measurements presented?
- Photobleaching appears to be a significant factor (Fig. S1). It would be helpful to discuss this in the main text, with supplementary details cited as needed. Additionally, clarification of the rationale behind the initial bleaching durations (5 s or 30 s, as shown in Fig. S2) would enhance the discussion.

[Applications]

- The rationale and advantage of spatial imaging using gradient fields are not immediately clear. A quantitative analysis considering signal strength and fluorescence background noise, as well as a comparison of imaging resolution and penetration depth with conventional direct fluorescence imaging, would be helpful to evaluate the feasibility and applications of this idea.
- While sensing molecular motion via protein interactions is an interesting concept, its advantage over optical measurement is not convincingly demonstrated. Polarization scattering becomes significant primarily in deeper tissues, where the relatively lower signal strength of MFE and ODMR compared to direct optical measurements is likely to be a limiting factor. A quantitative comparison would help clarify the feasibility and potential benefits of this approach.
- In the abstract, the following sentences are speculative and not directly supported by the results presented in the manuscript: "Magnetic resonance measurements using fluorescent proteins enable novel technologies; examples include 3D spatial localization of fluorescence signals using gradient fields (i.e., Magnetic Resonance Imaging using a genetically encoded probe), and sensing of molecular motion or protein interactions, without the limitation of light scattering in tissues. MFPs also enable a number of applications through their MFE." These claims should be either simplified or removed from the abstract to be more consistent with the demonstrated findings.

Referee #3

(Remarks to the Author)

This is an interesting manuscript developing proteins with the LOV2 domain with electron transfer to a flavin type molecule to internally generate a triplet state upon optical excitation in the blue. This triplet can then be detected in biological systems (specifically when expressed in bacterial cells) showing magnetic field effects (MFE) and optically detected magnetic resonance (ODMR). The systems are evolved for improved MFEs, and the characterization is very nice, but the overall sensitivity is very low and the concept of "quantum biological sensor" appears overhyped. While the underlying science is interesting, the magnetic fields needed to improve upon existing technologies for any of the proposed sensing schemes seem far too unrealistic. At the very least, I'd really want the authors to show a realistic path to improved measurements using their labels, not just "potential applications" that based on their data seem far, far out of reach. This is a very nice proof-of-concept, with very nice descriptions of what is being observed for the lay reader, but there just doesn't seem to be enough impact. I could possibly be convinced otherwise, but it would have to have much better s/n for much lower numbers of proteins. Or a much more impactful and realistic target for sensing. Specific comments are below.

All of the below comments are deemed major (except for the comments praising their work, as I do like the basic science, ideas, and characterization)

The abstract is rather vague about applications and details - ODMR detected from proteins at room temperature, but it is of the protein that is put into the cell, not of other species in the cell. There is no spatial information obtainable, and Magnetic field gradients needed for subcellular info are likely to be enormous. MFE is not defined even though it is presented in the abstract. The authors claim "MFPs for multiplexing or lock-in amplification of fluorescence signals, opening a new approach to combining or extracting multiple signals from a biological measurement." This has been done for fluorescence - see for example A.E. Jablonski, et al, J. Phys. Chem Lett (2012) or JACS (2013), and probably later papers. Not for magnetically modulated proteins of course, but still, similar.

Please define (or actually remove) "quantum sensing". You're sensing magnetic fields or spin or molecular interactions. Be specific. Using spin, sure, that's quantum, but so is fluorescence or absorption or many other modalities. All depends on definition and perspective.

Fig. 1: Experimental details are needed.

how much protein is being probed? signal in D is strong, so that's good. but odmr signal (E) from single cell (C) is very weak. Are C,D,E signals all from the same sample?

The signal from a whole cell (c) is total fluorescence and it is modulated by 30% with magnetic field changing from 0 to 10 mT.

Is there no photobleaching? Signal seems relatively constant at zero level, but the contrast decreases to -27% or so at long times. is this significant and from what does this arise?

The modulation (D) seems quite strong, which is great, but why is the S/N in E so low? Shouldn't the amplitude at the fundamental frequency be more like 15% in E or is there modulation at all frequencies and only a very slight increase in modulation depth at the 600MHz resonance that is presented in 1E? Why are the Fourier amplitudes not matching from D to E?

P. 3, bottom: What will ODMR be used for? you're detecting either the protein only or local magnetic fields, not species around it, correct?

P. 4, multiple potential sensing modalities are suggested. sorry, but, show something realistic here that is supported by your data. S/N is low for whole cell ODMR (Fig 1E). Lots of protons around, metals, etc. magnetic field effects, sure, but what, specifically can this be used for? Where is protein in cell? is it attached to a specific protein as a fusion? are you proposing sensing radicals just floating around in the cytosol or is there some specific interaction that can be sensed when this is expressed as a fusion protein? What is the intracellular concentration needed and what signal would be observable? All of this becomes a concern because the total odmr signal from your entire cell is very low in fig. 1E.

For spatial imaging, what is the sensitivity of ODMR for fusion proteins? Will fluorescence differences and MFE sensitivities be sufficient such that this can be detected with higher resolution than just straight fluorescence? What is the net benefit here? Is it realistic to have magnetic field gradients that are sufficiently strong that MR contrast can be seen across the scale of a single cell and improve spatial resolution relative to fluorescence alone? The gradient across a 500 nm distance of >10 mT seems more than a bit challenging to achieve. Please comment.

These are interesting ideas, but if they are not practical, then it is unclear what the innovation is that warrants publication in Nature at this time.

P. 5: specifically figure 2: Overall, experimental details are lacking - detection volumes, excitation intensities, imaging parameters, etc.

I'll believe that these are hyperfine peaks, which is pretty cool, but the sampling seems pretty close to the limit for resolving these, and the data is quite noisy. This looks promising, but do the sidebands change frequency with applied magnetic field? or is the s/n too low and sampling time insufficient to determine this?

The authors indicate that they need negative control cells to subtract signal off. So, an internal reference is needed as with G. Marriott's Optical lock-in detection (OLID) approach to fluorescence-based recovery? If so, this is more complicated than not needing an additional reference that must be subtracted. It is claimed that this is due to instrumental artefact/issues so without optical modulation it may be difficult to get rid of this modulated background. 2D suggests that other proteins/background emitters or species only give a small non-zero Delta I/I signal, or is this zero consistent with noise? Straight modulation of signal by varying the magnetic field should be able to give direct lock in detection without a negative control as the reference. Is there a path to fix this or will this always be an unavoidable problem? How far was the magnet z-position moved to change the fields? what precision is needed for improving resolution or other applications? is this achievable?

Fig 2A: an entire continuous carpet of cells is detected here. what is the spot size and estimated number of cells for each of the measurements in panel 2E? what is the integration time for the data presented in B, C, E? To improve resolution, you'd need a field gradient of ~ 5mT/500nm, correct (does this really mean a 10,000 T field needs to be applied for improved spatial resolution)? Is this in any way achievable or are my estimates from data presented incorrect? I'd really like to see some actual application or a realistic proposed application based on data presented, not just saying we have a quantum sensor. Don't get me wrong, the data and innovation is quite cool, but it is unclear how this actually leads to anything

biologically meaningful. I'm not saying it won't, I'm just not convinced by the hypothetical arguments the authors make based on the data presented.

P. 6: The description of where signal comes from (molecularly and quantum mechanically from the triplet states and their connections with RF and optical excitation is excellent. In contrast to the later statement of an artefact from the experimental system, this description suggests that no internal reference is needed and modulation of the magnetic field results in modulation of fluorescence signal as in fig 1. Only later do the authors say this is because of the instrumental issues. this is important, as background subtraction using negative control cells to see signals in 2E is at best inconvenient, and at worst disqualifying for many applications. Can modulation with lock-in detection not automatically remove this unmodulated background? or does residual background modulation reduce contrast? Are there specific parameters of the excitation or applied magnetic fields that could enable your label to be recovered without any background interference other than shot noise?

P. 7: only here is the instrumental artefact mentioned that needs to be subtracted. I appreciate the transparency, but this should be earlier as the reader is likely to be confused already about the need for negative control background subtraction. Further, in Fig. 2: the subtraction to yield hyperfine peak contrast from 2B to 2C is suggestive, but the noise is very high. The hyperfine side bands should not be present in the instrument artefact, correct? can you use a different modulation scheme (frequency modulation/heterodyne detection) to more directly recover your signal of interest without requiring a negative control cell population for subtraction?

P. 8: Characterization of species involved is excellent. Poor S/N and vague (and seemingly unrealistic) discussion of sensing applications, however, lessen impact as presented. the science and system are exciting though.

P. 9: the evolution/genetic engineering for MFE is also very nice.

P. 11: Fig 5 caption: Jablonski, et al as referenced above did this separation of two fluorescent proteins based on modulation. there are probably other examples of this from the same and other groups at this point. They used a second laser to depopulate the dark/triplet state, then could use lock-in amplification (i.e. a fourier transform) to decouple different species. EGFP, AcGFP.

in 5 B,C, Occurrences might be better for y-axis than frequency. frequency is used elsewhere so was initially confusing. Also, tau on and off are not explained in the caption at all, so figure does not stand on its own. Even in the text, are tau on/off defined? presumably the magnetic field is modulated and the rise and decay are measured or the period between B field on times is modulated to get these decay times? this needs to be described in more detail. How are the two populations separated when mixed? Please provide experimental parameters (imaging time, modulation parameters, intensities, detection volumes, how demodulation is done, time traces vs frequency responses, etc)

P. 13: what "quantum biological sensing"? robust quantum measurements? sure, spin is quantum, but so is fluorescence. I don't see what is really sensed here as the s/n and resolution are both lacking on anything close to a single cell, let alone subcellular info.

The science overall is very nice, as is the characterization and evolution, but the "quantum sensing" are too far into the potential regime, and an actual sensing system should probably be shown or at least rationalized based on real data. Basically, the probe is being detected, with no real sensing occurring. Such sensing should improve on what is currently possible (or show a path to doing so), without overselling the potential applications - especially when the sensitivity or suggested potential seems too low.

I will be delighted to see data that shows otherwise, but I'm not sure the technology is there yet, or will ever be there (I hope I am wrong about this, but it is the authors' responsibility to demonstrate this).

their imaging system generates (large) fields of ~ 1 T/m. not sufficient for intracellular or even cellular resolution based on ODMR. I don't have a good sense of what a realistic application would be that would advance the field - please explain if going beyond demonstration of ODMR.

tumbling rate for binding partners. maybe. but this would be for LOTS of proteins in solution due to sensitivity. this is ok, but not sure this can be done in cells or in real conditions. What are the timescales that can be probed and how does this compare with fluorescence anisotropy (and extensions of FA to larger complexes)? in vivo tumbling to measure interacting partners would be interesting, but again, need a very large sample, and fluorescence still must be detected (at 450 nm, no less, so in vivo applications are not viable).

Version 1:

Reviewer comments:

Referee #1

(Remarks to the Author)

The authors did a great job responding to reviewer comments. Congrats on a magnificent paper!

Referee #2

(Remarks to the Author)

The main change in the revised manuscript by Abrahams et al. is the inclusion of two new experimental datasets that demonstrate proof of concept for spatial localization and microenvironmental sensing. These additions address the common critique from all three referees regarding the lack of convincing application data.

Spatial resolution in bulk cell samples is now demonstrated. However, the practical utility of this technique remains unclear. The reported spatial resolution is on the order of 5–10 mm, likely limited by both spin resonance linewidth and signal-to-noise ratio. While magnetic fields can penetrate deep tissues, the fluorescence detection on which this method relies typically has a penetration depth of only a few millimeters, even under diffusive conditions.

The magnetic field sensing results in Fig. 6 confirm that the method is sensitive to the local microenvironment. The proof-of-concept results are interesting and may inspire readers to consider new directions for biomedical sensing. Nonetheless, the current sensitivity remains several orders of magnitude lower than that of NV centers. Moreover, the reported integration time of 200-500 ms per point is relatively slow for in vivo sensing or imaging applications.

Since this revised version was submitted, Reference [50] by Feder et al. has been published in Nature (<https://www.nature.com/articles/s41586-025-09417-w>). Given the relevance in principle and scope, it would be appropriate to cite this paper in the Introduction, revise the discussion accordingly, and briefly compare the two techniques—particularly in terms of signal-to-noise ratio and measurement speed.

According to the author guidelines, the first paragraph should contain references (as in the original submission). The citation format of Reference 9 is also incomplete.

In Fig. 5B, please display the grayscale map.

The methods used to obtain Figs. 5E and 5F are not clearly explained. Was the z-position determined by the frequency of the B_1 field? If so, it would be clearer to plot the data with frequency as the primary x-axis and the corresponding z-coordinate as a secondary (top) x-axis. The simultaneous measurements in Fig. 5F appear to exhibit higher noise (or fluctuations) and greater crosstalk between the two samples; please comment on it. There is also little description of the deconvolution method applied in Figs. 5E and 5F. What was the functional form of the point spread function? It is somewhat surprising that a simple deconvolution produced two clearly separated peaks from such noisy data.

In Fig. 6A, is the solid curve a theoretical fit using the function $(ax + b)^{-1} + c$? Please specify the best-fit parameters and compare them with theoretical expectations.

Quantum Spin Resonance in Engineered Magneto-Sensitive Fluorescent Proteins Enables Multi-Modal Sensing in Living Cells
Abrahams et al.

Response to Reviewers

Referee #1 (Remarks to the Author):

1.1. The manuscript by Abrahams et al describes the physico-chemical mechanism underlying the response of flavin-associated fluorescent proteins to magnetic fields, demonstrates the use of directed evolution to alter this response, and illustrates potential applications in multiplexed imaging and lock-in amplification. Overall, this is an important and fascinating study that will no doubt garner interest from a broad audience. The overall story and data are convincing. I just have a few suggestions for the authors.

We thank the reviewer for their highly positive appraisal of our work, as well as the detailed feedback which we address point-by-point below.

1.2. The claim that the R11f mutant has a faster saturation rate is not very evident based on the data shown in Fig. 4. It would be great to see some fitting or other quantification.

We thank the reviewer for identifying this mismatch. Indeed in the *majority* of the data presented previously this was the case e.g. (wide-field microscopy images captured at high illumination powers), though in some measurements (e.g. cuvette-based characterisation) the rates were different. We have since dug into the details of this and found that the primary driver of the R11f (now named MagLOV 2 fast) mutant not having a faster saturation rate in some cases was due to high LED powers used for microscopy experiments which essentially saturated these rates. This is in contrast with the conditions under which selection was performed (for instance, see the bulk liquid data SI-Fig11 presented previously in which MagLOV 2 fast is clearly faster). In the updated manuscript we have re-collected all of the relevant microscopy data with lower LED powers and imaging parameters, which now show an excellent correspondence between the different measurement modalities (e.g. compare Main Text Fig2 with Supplementary Figure S11). As the reviewer suggested we have also now fit rate functions to all of these data, and the relevant figures (as above) now present saturation timescale parameters so they can be directly compared. This has also been done for ODMR data.

1.3. The authors observe that the ODMR response is much larger (as a %) in single cells than it is in the bulk. The authors attribute this to “increased background fluorescence, especially in the case of cell conglomerates”. Can you be more specific? Autofluorescence of what? Why should it be different in a more concentrated solution of the same thing? Perhaps there are more complex (and interesting) phenomena at play, and this % change seems like a significant practical issue.

In summary, in response to this reviewer comment we have new discussion of this measurement in Supplementary Section S1.9, have moved this bulk data in its entirety to the Supplementary Material where we feel it is better suited as an additional control measurement, and have included additional experiments that explore how measured contrast values depend on measurement and sample parameters.

The bulk situation we describe (now in Section S1.9) is a conglomerate of cells scraped from an Agar pad. As such it is expected to contain a mixture of cells at high concentration, cellular debris, and Agarose/LB debris which is highly auto fluorescent – together these factors produce the fluorescent background, and as is now explained in S1.9. This bulk/conglomerate method was originally used because it was the simplest way to quickly determine if our samples exhibited any response before performing more careful quantification (i.e. subsequent measurements using microfluidics, a dilution to single cells on a coverslip, and now purified protein). When performing directed evolution, given the numerous colonies to screen, this is a fast method albeit not the cleanest. The significance of the bulk cell sample experiments as presented in the previous manuscript was simply as control experiment to ensure that there *is* an ODMR signal originating from the cells (confirming our hypothesis, but nonetheless the claimed discovery as this had not been observed prior).

To further address the question of why the percentage contrast may change between different samples or measurement modalities our updated manuscript includes a new Supplementary Section S1.19. Here we performed characterisation of MFEs for multiple samples (within cells) with different LED parameters highlighting how this leads to differences in dynamic response (which may then be anticipated between different measurement types). Furthermore, we have added a new section S1.12 on quantifying sensitivity of our measurements in response to other review comments; here a new Fig S6 explores how ODMR contrast (for purified protein, where we can better measure absolute concentrations) varies depending upon imaging and concentration parameters – which similarly differ between our bulk cell microscope samples (Now Fig S4) versus single cell microscope samples.

1.4. The authors speculate about potential applications to MRI-like spatial readout of fluorescence and actuation rather than imaging. I think it's fine to have such

speculations. However, in my opinion, anything that is not actually demonstrated in the study belongs in the discussion section and not the abstract or introduction.

To address this comment, we have now undertaken a significant new body of work which involved building a novel fluorescence-based MRI instrument (explained in detail in new Supplementary Section S1.20) and demonstrate its application to spatial readout of fluorescence over large spatial volumes. This is presented in the new Section 5.2 and Figure 5 of the updated manuscript. As such we have strengthened statements about possible applications to make clear such a measurement is possible.

Meanwhile, as suggested by the reviewer we have moved discussion of e.g. tumbling sensing and possible future development of magnetogenetic actuators to the discussion section.

1.5. Speaking of speculation (which I admit is fun!), would an MRI-like readout be limited to 1 dimension or are there prospects for encoding additional dimensions via a pulse sequence analogous to what is done in MRI? Are there fundamental physical limitations on lifetimes, etc, that would make this difficult?

Fluorescence MRI is not limited to 1 dimension; it can be encoded in 3 dimensions even without using pulse sequences (though using pulse sequences is an interesting avenue for future research) by creating a series of different magnetic field gradients.

One approach to this would be to have a multiple coil system that allows magnetic/RF field gradients to be created and dynamically rotated in 3D to scan the resonance condition across a sample. Similarly, in cases where there is a single fluorescence target to be localised in a 3D volume this could be done by rotating/moving the sample using the exact same instrument as we present in Fig 5 – for example, a sample would be measured three times in each of three orthogonal orientations to localise a fluorescent reporter in 3D.

In terms of the fundamental physical limitations of such a measurement, these would include the photostability of the MFP and the sensitivity of the instrument. We now discuss these limitations – and how overcoming them would motivate further development of MagLOV proteins specialised for this application - in the new Supplementary Section 1.20.3.

1.6. Also on speculation, it is not immediately clear to me how the mechanism described here could be used for protein actuation, since it requires the build-up of states through a fluorescence cycling process, if I understood it correctly.

Indeed, while this is an exciting prospect it is not yet clear how best to achieve this in practice, hence we have limited any discussion of this to a passing remark in a list of

possible future scenarios. If we *may* speculate here, we envision a charge state transfer from MagLOV (or similar) to a fused protein that would cause it to switch states (e.g. to impact binding/confirmation or other functionality), and that MagLOV itself might be excited using BRET/FRET charge transfer from a bioluminescent protein – a similar philosophy to recent studies developing novel optogenetic actuators [1].

[1] <https://www.biorxiv.org/content/10.1101/2023.06.26.545546v2.full>

Referee #2 (Remarks to the Author):

2.1. This manuscript by Abrahams et al. presents the development of magneto-sensitive fluorescent proteins based on spin-correlated radical pairs with flavin molecules. The engineered proteins exhibit enhanced magnetic field effects (MFE) in response to a static magnetic field (10–30 mT) and demonstrate improved optically detected magnetic resonance (ODMR) characteristics under an alternating magnetic field (~0.3 mT) at resonance frequencies corresponding to hyperspectral energy level splitting. Notably, the proteins achieve an intensity contrast as high as 30% in solution (Fig. 1D), 4% in a single bacterium (Fig. 2E), and 0.05% in bacterial culture. The manuscript also explores the magnetic sensitivity of several genetic variants and successfully demonstrates multiplexed detection of two distinct fluorescent proteins. Potential applications, such as spatial localization, local magnetic noise sensing, and protein binding assays, are discussed.

2.2. Overall, the approach is technical sound. The observed enhancement of MFE and ODMR effects in live cells is particularly noteworthy. However, the manuscript lacks a quantitative comparison of sensitivity (contrast) relative to other organic molecules and inorganic nanocrystals, making it challenging to assess the level of innovation.

We thank the reviewer for this suggestion. In response we have now performed a range of additional experiments and present these in the new Supplementary Section S1.12. In particular, we now present data and calculations of sensitivity for purified samples of our protein, we compare this to sensitivities presented in other recent studies as well as inorganic nanocrystals (i.e. NV nanodiamonds as they have the best sensitivities), and we also include the new Table S2 summarising other relevant parameters (e.g. contrast % and ODMR linewidth) for our proteins and those from recent studies. In this new supplementary section, and the updated main text Discussion, we also clarify that the key innovation of our method compared to inorganic nanocrystals is that it is an *entirely biological*. This allows it to be genetically encoded and engineered with biotechnological methodologies (as done in our work), and it can be produced and measured *in situ* in living cells (also done in our work) - these advantages open new and unique application possibilities (e.g. genetically engineered tissue dependent expression) when compared to methods such as NV nanodiamonds.

2.3. From an application standpoint, the feasibility and sensitivity of the magnetic resonance-based approach, as well as its advantages over existing methods, remain unclear. The figures could be improved to enhance clarity and visual presentation.

In response to this suggestion (similar to Reviewer 1 comment 1.4) we have now undertaken a significant new body of work which involved building a novel

fluorescence-based MRI instrument (explained in detail in new Supplementary Section S1.20) and demonstrate its application to spatial localisation of fluorescence over large spatial volumes. This is presented in the new Section 5.2 and Figure 5 of the updated manuscript. We believe this concretely demonstrates the feasibility of magnetic resonance based localisation of signals, though of course our instrument and technology only represents an initial proof-of-concept with large room for improvement.

We have also, throughout the manuscript, significantly improved explanations and figure captions and clarity of data presentation.

Questions and Comments:

[Mechanisms]

2.4. Fig. S6 could replace Fig. 1B and be cited in the second paragraph of Page 6.

Fig 1B was intended to give readers a quick and graphic overview of the mechanism, namely relating fluorescence to the radical pair dynamics. We believe it is useful to have significantly simplified diagram such as this early in the paper so that generalist readers can quickly grasp the broad idea of the work, even if the more detailed analysis is beyond their interest or disciplinary background. Nevertheless, we have moved the proposed mechanism schematic which was formerly Fig S6 to be part of the updated Fig 3 as it then better fits within the expansion of the mechanistic discussion (please see below).

This being said, we are open to feedback from the reviewers and editor about the clarity of presentation of these schematics and how to best trade-off general audience readability with technical depth.

2.5. What are the typical values of key transition rates—fluorescence, intersystem crossing (ISC), back electron transfer rate (k_{BET}), and reaction rates (k_{A} , k_{D})—for the molecules studied in this work?

As above, we have now included a tentative photoscheme labelling these processes in Figure 3. However, without extensive further work, we can so far confirm only that (i) the emission stems from FMN, (ii) the SCRPs are triplet born, and (iii) the semi-quinone is formed as a stable radical at later times, and can potentially be transformed to the fully reduced form (FMNH⁻) upon extensive illumination. Hence, the later species (FMNH⁻) is unlikely to be relevant to the MFE mechanism here. The identity of the radical pair species and the counter radical is hypothesised based on previous work on AsLOV2 C450A [Richter 2005, Eisenreich 2008]. Approximate rates based on related model systems are given in Table 6.2 below.

Rate	Process	Value	Source
ISC	Intersystem crossing	$3e8 \text{ s}^{-1}$	AsLOV2 WT, Kennis 2003
ET	Electron transfer	$3e5 \text{ s}^{-1}$	FMN+AscH2, Evans 2016
BET (k_S in simulations)	Back-electron transfer	$8e4 \text{ s}^{-1}$	FMN+AscH2, Evans 2016
H/Dep (k_f in simulations)	Protonation/Deprotonation	$1e6 \text{ s}^{-1}$	FMN+AscH2, Evans 2016
k_{FMNH}	Return (ox) to FMN GS	$3e3 \text{ s}^{-1}$	FMN+AscH2, Evans 2016
k_D	Return (red) to TRP GS	$> k_{FMNH}$	as MFE enhanced (Kattnig 2016)

Future *experimental* work would be able to pursue spectroscopic measurements (ns-broadband transient absorption with MFE probing) for purified MagLOV variants, which could further investigate the proposed photoscheme and determine the relevant rates in MagLOV. Nevertheless, to provide initial estimates for these parameters we have now performed spin dynamics simulations, based on literature values and fitting to the data we measured, which are presented in the new Supplementary Section SI1.18.

References

- Richter, G., Weber, S., Römisch, W., Bacher, A., Fischer, M., Eisenreich, W.: Photochemically Induced Dynamic Nuclear Polarization in a C450A Mutant of the LOV2 Domain of the Avena sativa Blue-Light Receptor Phototropin. *Journal of the American Chemical Society* 127(49), 17245–17252 (2005).
- Eisenreich, W., Joshi, M., Weber, S., Bacher, A., Fischer, M.: Natural Abundance Solution ^{13}C NMR Studies of a Phototropin with Photoinduced Polarization. *Journal of the American Chemical Society* 130(41), 13544–13545 (2008).
- Song, S.-H. et al. Absorption and emission spectroscopic characterisation of combined wildtype LOV1–LOV2 domain of phot from *Chlamydomonas reinhardtii*. *Journal of Photochemistry and Photobiology B: Biology* 81, 55–65 (Oct. 2005).
- Kennis, J. T. M. et al. Primary Reactions of the LOV2 Domain of Phototropin, a Plant Blue-Light Photoreceptor. *en. Biochemistry* 42, 3385–3392 (2003).
- Evans, E. W. et al. Sub-millitesla magnetic field effects on the recombination reaction of flavin and ascorbic acid radicals. *en. The Journal of Chemical Physics* 145, 085101 (2016).
- Kattnig, D. R. et al. Chemical amplification of magnetic field effects relevant to avian magnetoreception. *en. Nature Chemistry* 8, 384–391 (2016).

2.6. How do the MRE and ODMR signal strengths ($\Delta\%$) vary as a function of the static magnetic field strength (B_0)?

Following this suggestion we have now performed a MARY experiment (MFE as a function of applied magnetic field strength) on the purified solution of MagLOV 2 fast,

which is presented in the new Supplementary Section S1.18.2. We were also able to simulate the data successfully with the assumption of a [FMN.- TrpH.+] radical pair, which is now presented in the new Supplementary Section S1.19.

For ODMR, we *do* measure contrast values at a range of static B₀ values (e.g. Fig 1E). However, with our existing instrument is challenging to quantitatively compare these in terms of absolute contrast % due to the fact that our antenna response (i.e. magnitude of delivered B₁ field) varies as radiofrequency changes, which is now characterised and explained in the new Supplementary Section S1.13.1 and Fig S7. In process of reviewing our manuscript we developed ~10 additional RF antenna geometries to try to make this response as uniform as possible, as well as the calibration protocol outlined in Fig S7. In addition to the above, we were unable to perform ODMR measurements at significantly different B₀ fields due to our RF amplifier having a limited operational frequency range for the delivered B₁ field (which necessarily constrains the explorable B₀ values according to the resonance condition).

Nevertheless, developing novel RF antennae and acquisition of different frequency-range amplifiers to explore ODMR at significantly different frequencies (and hence B₀) would be an interesting direction for future work.

2.7. Providing theoretical equations for I₂ and I₃ (in Fig. S6) as functions of B₀ and B₁ would be beneficial. Additionally, a direct comparison between theoretical predictions and experimentally observed sensitivities would strengthen the analysis.

The theoretical underpinning of the spin-radical pair system is complex. To perform theoretical predictions, one would require a detailed understanding of every species involved in the photocycle, which thus far have not been fully elucidated. Nevertheless, we have made substantial progress in determining these species and continue to do so (see Main Text Section 3 and new Supplementary Section S1.18 for characterisation). Furthermore, following this suggestion we have performed simulations based on our current understanding of the system which fit the data for both ODMR and MARY experiments. We describe the theory behind these simulations and the assumptions needed to perform them, and present these results, in SI S1.19 and Fig. S12 B.

We hope that future work, e.g. spectroscopic studies with both optical cavity and EPR methods, would be able to clarify the identity of the counter radical(s) and probe the steps directly involved in SCRP formation in MagLOV.

[Signal Contrast and Sensitivity]

2.8. The manuscript reports an ODMR contrast of 4% from a single live cell in culture (Fig. 1E), where the number of proteins is estimated to be at most 10⁵ per cell. This sensitivity is described as three orders of magnitude better than current electron spin

resonance (ESR) techniques. Could the authors provide additional data and a statistical analysis across different single cells to confirm this improvement?

Following the reviewer's suggestion we have performed extensive single cell characterization and updated all the characterisation data in the main text to be from this new data set. In a single field of view there are roughly 500-1000 distinct cells. We have updated Figure 1 to now display the full statistics of these measurements: in panel D we process the data by individually fitting the fluorescence curve of each cell in the field of view to get the background-subtracted ODMR signal, and present the mean, standard deviation, and raw data from a representative single cell.

Furthermore, we have now written and include a new Supplementary Section in S1.12 we which estimate the per cell B_0 sensitivity, and also the concentration normalised B_0 sensitivity based on new data we collected for purified solutions of MavLOV. This section also compares our calculated sensitivity to those of other techniques (e.g. NV Nanodiamonds), and makes that one of the key advantages of our method (compared to other ESR methods) is that it is an entirely biological system suitable for engineerable production and measurement in living cells.

2.9. The contrast of 4% in single cells decreases to 0.05% in cell conglomerates (Fig. 2B), attributed to increased background fluorescence. What is the autofluorescence intensity (e.g., photons per square micrometer) in multi-cell conglomerates compared to isolated single cells? Further investigation into the underlying reasons for this contrast reduction would be valuable. Are the cell conglomerates a monolayer or multi cell layers?

In summary, as outlined below and in response to Review 1 point 1.3, the single cell data is a much cleaner, and quantifiable way to present our findings, hence we have moved the cell conglomerate data to the SI.

The multi-cell conglomerates were colonies scraped onto a slide and sandwiched with a coverslip, as such likely not a perfect a monolayer. These measurements are useful in that they allowed us to quickly verify that the ODMR signal required genetic expression of MagLOV, but they are not as effective for quantitative characterisation of MagLOV due to the way in the measurements were performed: colonies were scraped directly from agar plates containing highly autofluorescent media (LB) and imaged. These limitations are now explained in the new Supplementary Section S1.9.

To address the reviewer's request for quantification of autofluorescence; under 450nm, 1.5 W/cm² illumination conditions, in Fig. S4 A at frame zero, averaged over the MagLOV region the brightness is 486 photons/s/um², and averaged over the pTU1 region it is 443 photons/s/um². By contrast, for single cells expressing MagLOV under the same illumination conditions were observed to have a brightness of 30 photons/s/um²,

highlighting the significant (uncertain) contribution of the depth of cells and other fluorescence sources (e.g. LB-agar) which together contribute to the uncertainty in measurements with the “colony scrape” method – hence their de-prioritisation to the supplementary.

Furthermore, from the single cell data in Fig S10 (albeit under different conditions) the upper bound on autofluorescence is orders of magnitude smaller than that of the fluorescence due to MagLOV, also consistent with the finding that conglomerate measurements are heavily impacted by factors such as the unknown thickness of the layer, and the presence of non-cellular matter in the mix (e.g. cellular debris, and agar containing highly auto-fluorescent Lennox broth – these colonies were not washed prior to imaging).

To further investigate the underlying reasons for contrast changes in different measurements, we have also included S1.19 that explores MFE contrast and saturation rate at different LED powers.

2.10. Considering potential magnetic-field sensing, what would be the detection sensitivity values of static (B_0) and oscillating (B_1) magnetic fields, expressed, for example, in $\text{mT}/\sqrt{\text{Hz}}$?

This is a good question and is now addressed in a new Supplementary Section S1.12 “Magnetic Sensitivity”. The sensitivity is dependent on the measurement context, the variant measured, etc. We have estimated the sensitivity for both MFE and ODMR, in both cases measuring the sensitivity with respect to the static magnetic field B_0 , which we feel is appropriate – clearly for MFE, and for ODMR because any sensing application involves measuring a change in the spectra, i.e. how the resonance curve shifts with respect to B_0 (this is also the standard measure e.g. for NV-sensing). These factors are discussed in the new Section S1.12, including the detailed calculations as well as comparison to similar systems in the literature.

2.11 How do the measured contrast values compare with previous studies on ODMR and MRE using organic molecules and inorganic materials?

We address this in the Supplementary Section S1.12 which includes a new Table S2 summarising contrast values from previous studies and compares these our work; this includes both biological systems as well as NV nanodiamonds.

2.12 What are the optical illumination intensity levels experimentally used for different samples?

This data is now more clearly represented in the SI, for instance in Widefield Imaging Protocols, Spectroscopic characterization and Spatial Localisation Experiments. To

investigate the impact of illumination intensity we have also collected new data which compares the ODMR response of purified protein samples at different illumination intensities (Table S1), and a new Supplementary Section S1.19 which explores how illumination level impacts contrast/rate of magnetic field effects measured. Finally, to improve presentation clarity, we have also added a new Supplementary Table S3 which summarises the illumination intensities used for experiments in the paper.

2.13 What are the signal integration times for the different measurements presented?

Imaging integration time and acquisition protocols are clarified in the updated Supplementary Section S1.7 and e.g. Fig S2. For example, for all widefield single cell resolved measurements, exposure time was 200 ms per point. For MFE measurements imaging during magnet on/off periods allows collection of the dynamic response waveforms, and these are averaged over five on-off periods to give the resultant data (e.g. as in main text Fig 2). This process of image collection, processing, and subsequent averaging over multiple measurement periods is now presented in detail in Supplementary Section S1.15.2 and shown in new Fig. S10. For ODMR sweeps, we scan from high to low frequency, at each frequency the LED and RF is on for 250 ms and the camera exposes for the final 200 ms, followed by 250 ms of darkness (and RF off). A full sweep of 100 points therefore takes 50 seconds. We did not average sequential sweeps together so 50 seconds is the full acquisition time.

2.14 Photobleaching appears to be a significant factor (Fig. S1). It would be helpful to discuss this in the main text, with supplementary details cited as needed. Additionally, clarification of the rationale behind the initial bleaching durations (5 s or 30 s, as shown in Fig. S2) would enhance the discussion.

Some photobleaching is inevitable, as a fraction of the molecules gets stuck at the flavin semi-quinone state (FMNH.), which only slowly recovers to the ground state. However, this reduced state is NOT the 'source' of the MFE as is the case for example with (Xiang, 2025). From the raw data in Fig S11B one can see the MFE is present from the start. However, it is worth pointing out the degree of photobleaching in MagLOV is minute in comparison to other flavin-based systems reported in the literature (e.g. FMN+HEWL in Dejean, 2020), which makes MagLOV a particularly attractive molecule.

To address this suggestion, for all the single cell data, we now acquired the full time-trace (therefore the photobleaching is present in the raw data which could be processed by readers should it be of interest). We illustrate our processing method to account for photobleaching in the new Figure S9; here it is clear the initial photobleach gradient is sufficiently steep that it would severely impact the MFE step if included in averaging, hence that portion of the curve is excluded from further data processing, which we explain as the rationale of the bleaching duration chosen. Furthermore, in the new Fig S10 we show the background brightness (a combination of photobleaching and other

photodriven effects) is not monotonic decay; we now explain this rationale for use of a spline fit (rather than e.g. an exponential) to remove this background.

[Applications]

2.15 The rationale and advantage of spatial imaging using gradient fields are not immediately clear. A quantitative analysis considering signal strength and fluorescence background noise, as well as a comparison of imaging resolution and penetration depth with conventional direct fluorescence imaging, would be helpful to evaluate the feasibility and applications of this idea.

As suggested by the reviewer we have clarified the need for gradient field based methods for spatial imaging – in addition to *demonstrating* its feasibility this through developing such a system in the new Fig 5. In particular, imaging in scattering/absorbing tissues is a common challenge in biomedicine which has motivated development of a diverse range of methods to improve signal collection for such challenges, such as Fluorescence Molecular Tomography [1]. Similarly, to quote from [2]:

“Optical coherence tomography, photoacoustic microscopy, optical phase conjugation, and wavefront shaping allow imaging through scattering media, but these techniques have depth of penetration of 1 mm or less. MFI is in principle free from such limits, provided molecules with suitable fluorescence spectra and magnetic sensitivities are identified.”

To further address the reviewer’s question about comparison of our method to direct fluorescence imaging, we have added a new Supplementary Section S1.22 “Advantages of Lock-in Detection”. Here we perform simulations of depth-dependent Signal to Background Ratios (SBRs) suitable for fluorescence measurements in tissues to explore the typical practical limitations for fluorescence imaging in noisy/autofluorescence environments in biomedical applications [3]. This new section illustrates how lock-in can significantly improve signal strength, particularly in the presence of autofluorescence or other background signals, and outlines the practical limitations of such methods as well as explaining how they motivate directions for future engineering of MagLOV and related proteins for this family of applications.

[1] <https://www.mdpi.com/1999-4923/3/2/229>

[2] <https://opg.optica.org/oe/fulltext.cfm?uri=oe-18-25-25461&id=208324>

[3] <https://pubs.acs.org/doi/10.1021/ar400325y>

2.16 While sensing molecular motion via protein interactions is an interesting concept, its advantage over optical measurement is not convincingly demonstrated. Polarization scattering becomes significant primarily in deeper tissues, where the relatively lower

signal strength of MFE and ODMR compared to direct optical measurements is likely to be a limiting factor. A quantitative comparison would help clarify the feasibility and potential benefits of this approach.

As above, we believe this may be technically feasible however we have not yet performed experiments to investigate molecular motion. Since this is speculative, we have removed discussion of tumbling.

2.17 In the abstract, the following sentences are speculative and not directly supported by the results presented in the manuscript: “Magnetic resonance measurements using fluorescent proteins enable novel technologies; examples include 3D spatial localization of fluorescence signals using gradient fields (i.e., Magnetic Resonance Imaging using a genetically encoded probe), and sensing of molecular motion or protein interactions, without the limitation of light scattering in tissues. MFPs also enable a number of applications through their MFE.” These claims should be either simplified or removed from the abstract to be more consistent with the demonstrated findings.

As suggested by the reviewer, key statements in the Abstract/Introduction have been rewritten. In particular, we now state explicitly that spatial localisation *is* possible, because our manuscript *does* now directly support this through Fig 5. This is similarly true of statements of about quantum sensing of micro-environment, which are supported by the new Fig 6. In other cases which are *not* directly supported by our new data (e.g. relating to molecular motion) we have removed statements as requested by the reviewer.

Referee #3 (Remarks to the Author):

3.1. This is an interesting manuscript developing proteins with the LOV2 domain with electron transfer to a flavin type molecule to internally generate a triplet state upon optical excitation in the blue. This triplet can then be detected in biological systems (specifically when expressed in bacterial cells) showing magnetic field effects (MFE) and optically detected magnetic resonance (ODMR). The systems are evolved for improved MFEs, and the characterization is very nice, but the overall sensitivity is very low and the concept of “quantum biological sensor” appears overhyped. While the underlying science is interesting, the magnetic fields needed to improve upon existing technologies for any of the proposed sensing schemes seem far too unrealistic. At the very least, I’d really want the authors to show a realistic path to improved measurements using their labels, not just “potential applications” that based on their data seem far, far out of reach. This is a very nice proof-of-concept, with very nice descriptions of what is being observed for the lay reader, but there just doesn't seem to be enough impact. I could possibly be convinced otherwise, but it would have to have much better s/n for much lower numbers of proteins. Or a much more impactful and realistic target for sensing. Specific comments are below.

We thank the reviewer for their support for the general interest of our work and its underlying science. We also appreciate comments related to a desire for more proof of some of the applications discussed in our manuscript. In response we have performed a significant amount of new experimental work (including developing a new fluorescence MRI instrument) which adds proof-of-concept for several novel applications of our work. We also concretely define, and include a new application that demonstrates, quantum biological sensing as it is generally defined the field. Updates to the manuscript are numerous and so we will explain them in detail in response to the points raised below.

3.2. All of the below comments are deemed major (except for the comments praising their work, as I do like the basic science, ideas, and characterization)

3.3. The abstract is rather vague about applications and details - ODMR detected from proteins at room temperature, but it is of the protein that is put into the cell, not of other species in the cell. There is no spatial information obtainable, and Magnetic field gradients needed for subcellular info are likely to be enormous. MFE is not defined even though it is presented in the abstract. The authors claim "MFPs for multiplexing or lock-in amplification of fluorescence signals, opening a new approach to combining or extracting multiple signals from a biological measurement." This has been done for fluorescence - see for example A.E. Jablonski, et al, J. Phys. Chem Lett (2012) or JACS

(2013), and probably later papers. Not for magnetically modulated proteins of course, but still, similar.

We have addressed points raised here with several new applications (detailed at length below) as well as editing to clarify these statements. In particular:

- We now include ODMR data for *both* proteins expressed in cells (e.g. Fig2 main text), and purified proteins in solution (e.g. in new Supplementary Section S1.12).
- We clarify what is meant by *Spatial Localisation* in the manuscript, and include a new Fig 5 which shows how spatial localisation can be achieved on a 2D microscope slide and within a large volume using a novel Fluorescence MRI instrument we developed. This new application demonstration also makes clear that the length scales for spatial localisation we are targeting refer to the position of cells within a sample, rather than position of molecules within a cell.
- MFE is now defined when it first arises in the first paragraph of the Introduction.
- We agree that there exist *other* methods for multiplexing or lock-in of fluorescent proteins (e.g. including usage of multiple reporters with different excitation/emission spectra), however we maintain that ours is a new dimension along which to multiplex (which could be multiplicatively combined with others). Furthermore, it is unique in its implementation (e.g. does not require the second laser and complex optical setup as in Jablonski *et al* 2012) making it ideal for a wide range of biotechnological assays. We have included reference to these past methods (such as Jablonski *et al* 2012 and Marriott *et al* 2008 discussed below) in the new Supplementary Section S1.22 which details the motivations and benefits of our method for lock-in signal detection.

3.4. Please define (or actually remove) “quantum sensing”. You’re sensing magnetic fields or spin or molecular interactions. Be specific. Using spin, sure, that’s quantum, but so is fluorescence or absorption or many other modalities. All depends on definition and perspective.

We agree with the reviewer that how a “quantum” system is defined requires some consideration [1]. In general, we believe the term “quantum sensing” as used in fields of e.g. NV nanodiamond sensing, typically covers measurements where the spin state is inferred from the measurement – and not simply cases in which a quantum phenomena (such as the photoelectric effect) involved. Here we are manipulating the spin state and reading out the relative proportion of spins in triplet/singlet state optically hence quantum sensing, and we clearly show this with the new Figure 6 where the local spin environment is sensed. To further clarify this point we have also added an explicit definition to the introduction: “quantum sensing tools (i.e. those whose function arises from spin-dependent processes)”. Nevertheless, we are very open to further refinement

of this definition if the reviewer/s or editor have opinions on how best to frame it within the broader field.

[1] <https://doi.org/10.1039/C9FD00049F>

3.5. Fig. 1: Experimental details are needed.

how much protein is being probed? signal in D is strong, so that's good. but odmr signal (E) from single cell (C) is very weak. Are C,D,E signals all from the same sample?

We have made a range of improvements to Figure 1 to address this suggestion. In particular, we have re-performed these measurements using optimised imaging parameters (e.g. lower LED power to reduce photobleaching) and MagLOV 2 which exhibits a higher MFE magnitude, and with a new radio antenna and RF system design which delivers high power that is more uniform across frequency. As a result the signal is much clearer in each panel, and we additionally include uncertainties in its magnitude via acquiring across 1000s of individual cells. Fig 1 C, D, E are all the same cell strain expressing the same protein with the same media/chemical environment, but they are not *literally* the same cells due to the desire for fresh samples for each measurement to reduce variability. These data are collected using the same imaging parameters, with the only difference being the use of an electromagnet in C for MFE measurements vs. permanent magnet and radio frequency generation in D, E (however in C, D and E the cells are situated on the radio antenna – simply turned off in C, to ensure the same lighting conditions). These setup parameters are detailed at length in the updated Supplementary including new sections (e.g. in S1.15) which go through step by step how data is collected and processed.

In terms of the question about how *much* protein is probed, the updated manuscript now also investigates the response of purified protein at defined concentrations (i.e. not in cells) in Supplementary Section S1.12, and we also performed experiments to quantify the intracellular protein concentration for biological samples as used in Fig 1. This is discussed in the new sensitivity calculations in S1.12 which presents sensitivity metrics for both purified protein as well as sensitivity per single cell.

3.6. The signal from a whole cell (c) is total fluorescence and it is modulated by 30% with magnetic field changing from 0 to 10 mT.

Is there no photobleaching? Signal seems relatively constant at zero level, but the contrast decreases to -27% or so at long times. is this significant and from what does this arise?

Please see response to reviewer 2 comment 2.14. In short, we have added a new Supplementary Section S1.15.2 which shows representative photobleaching curves

(e.g. Fig S9/S10) and explains in detail how we processed this to generate reproducible data e.g. motivating why we allow a delay for initial photobleaching of samples.

3.7. The modulation (D) seems quite strong, which is great, but why is the S/N in E so low? Shouldn't the amplitude at the fundamental frequency be more like 15% in E or is there modulation at all frequencies and only a very slight increase in modulation depth at the 600MHz resonance that is presented in 1E? Why are the Fourier amplitudes not matching from D to E?

The figure has been updated, old panel labels are denoted below by using underline.

Indeed, SNR in E was very low due the way we were imaging single cells. This is now much improved, thanks to a better LED intensity / exposure time settings, and improved radio power delivery due to the improved antenna. Hence in the updated figure this is clearer.

Regarding modulation depth and the former panels D and E; we note that E is not a Fourier transform of D. Panel D (now C) is an experiment where the magnet is switched on and off. Panel E (now D) is the ODMR spectrum in an entirely different measurement: a permanent magnet is present, and a radio wave signal is applied that drives the spin transition. The center frequency is determined by the electron spin resonance, this is now shown in the (new) Panel E: at different magnetic fields, the resonant frequency is shifted by 28 MHz/mT, indicative of an electron spin transition. We have clarified these experimental approaches with significant additions to the methodology explained in the Supplementary; for example Fig S2 outlines the sequencing of actions for the MFE versus ODMR measurements.

3.8. P. 3, bottom: What will ODMR be used for? you're detecting either the protein only or local magnetic fields, not species around it, correct?

This is a good point and we agree with the reviewer that our manuscript is now much stronger as we can show how this is useful through the multiple new applications we demonstrate. In particular, we have now included two proof of concept demonstrations: (1) localising the protein using an externally imposed magnetic field within a large volume using a novel Fluorescence MRI instrument as in Fig 5, and (2) sensing species around it (see Gadobutrol sensing and Fig 6). For instance, (1) could be extended to 3D localisation, and (2) could be extended to ROS sensing; these possibilities are now discussed in the Applications Section 5 of the main text, and in the updated Discussion.

3.9. P. 4, multiple potential sensing modalities are suggested. sorry, but, show something realistic here that is supported by your data. S/N is low for whole cell ODMR (Fig 1E). Lots of protons around, metals, etc. magnetic field effects, sure, but what, specifically can this be used for?

As outlined in response to the above points we have significantly expanded the demonstrated applications in our paper, and have added significant explanation and clarification of the utility of those previously included (e.g. multiplexing of multiple reporters, lock-in imaging for improved sensitivity as outlined in the new Supplementary Section S1.22) . Furthermore, we have updated the Discussion to make more precise how these techniques, alongside optimisation of MagLOV proteins and instrumentation, could be further tailored to realise sensing of other targets (e.g. ROS).

3.10. Where is protein in cell? is it attached to a specific protein as a fusion? are you proposing sensing radicals just floating around in the cytosol or is there some specific interaction that can be sensed when this is expressed as a fusion protein?

In our experiments the protein is not fused or tagged – it is expressed at high levels from multi-copy plasmids which are spread throughout the cell, and hence the protein is likely spread near-uniformly through the cell's interior. This is supported by our microscopy images of single cells (e.g. Fig 1D) which (admittedly at low resolution) show a spread of fluorescence distribution (and not, for example, accumulation at poles or other sub-locations).

The fact that the sensor *is* the protein means further developments such as tagging it to a cellular site of interest, or fusing it to another protein, are straight-forward possibilities achievable by either adding a binding tag to MagLOV, or co-expressing the two proteins as a single fused genetic construct. This is one of the key innovations, and advantages over related but not cellularly expressed sensors (organic molecules, or nitrogen vacancy nano-diamonds), in that our system can be integrated with other biomolecules with the same flexibility as other fluorescent markers (e.g. GFP) which are a cornerstone of the field.

Finally, in terms of *sensing* other molecules, we have not yet explored sensing of species within the cell, or spatial localisation of these proteins in the cell. However, our updated manuscript *does* demonstrate (in new Fig 6) sensing of molecular species with purified MagLOV (i.e. outside of a cell).

3.11. What is the intracellular concentration needed and what signal would be observable? All of this becomes a concern because the total odmr signal from your entire cell is very low in fig. 1E.

We thank the reviewer for the general suggestion for characterising the concentration of samples used throughout our work. Various changes and additions have been made to the manuscript.

For MFE data, we now quantified the intracellular concentration of MagLOV 2 expression (as used for sensitivity calculations in new Supplementary Section S1.12) to be ~10uM or approximate 10,000 copies per cell for data in Fig 1 and 2. This concentration gives a very strong signal. Furthermore, in Fig 4 we characterise single cells with very strong expression of MagLOV 2 (estimated from a genetic sequence expression predictor at ~200,000 copies per cell) in Panel B/C, and in Panel E/F measure a different genetic construct which has an intracellular expression level ~4,000 times lower (pVS-02-04, see Section S1.4) or approximately ~100 proteins per cell. This demonstrates the effectiveness of our system and e.g. the ability to measure the MFE effect in samples with many orders of magnitude of concentration difference.

For ODMR we also present quantification of relevant concentrations; main text data in Fig 1 and 2 is again at ~10uM or 10,000 copies per cell and (as discussed in points above) we have updated Fig 1D with better imaging parameters which exhibits much stronger signal. Meanwhile, in the new Supplementary Section S1.12 we *also* characterise ODMR traces for purified proteins at multiple concentrations.

The Gadobutrol sensing experiments were performed with a purified protein at a concentration of 3.7 uM, indicating that moderate intracellular concentrations (as above, we measure the intracellular protein concentration in the Fig 1 measurements to be ~10 uM, S1.12) are adequate for spin-based sensing.

3.12. For spatial imaging, what is the sensitivity of ODMR for fusion proteins? Will fluorescence differences and MFE sensitivities be sufficient such that this can be detected with higher resolution than just straight fluorescence? What is the net benefit here?

In response to this and other related comments we have made various additions to our paper demonstrating and analysing protein localisation techniques.

One factor of this question relates to the difference between standard fluorescence imaging and lock-in based methods for extracting signal from a noisy background, for example due to autofluorescence of a sample. These are discussed in detail in the new Supplementary Section S1.22. In short, in situations where background is large, straight fluorescence is highly limited as the “signal” cannot be distinguished from its surrounds. In contrast, by pulsing a magnetic field to perform “lock-in” measurements of MagLOV its fluorescence signal can be distinguished from background. We explore this in terms of the SBR (Signal to Background Ratio) in Section S1.22.

Another factor is the challenge of extracting signals arising at depth in scattering tissue or media (which may or may not be autofluorescent). Here, with standard fluorescence, light will scatter in the tissue meaning it arrives at a detector from multiple locations, and hence cannot be traced back to the source. Under certain circumstances, fluorescence modulated tomography (FMT) can be used to perform a reconstruction but requires a detailed structural model of the object being imaged *a-priori*. Overcoming these weaknesses, we show (in the new Fig 5) that our ODMR based localisation technique *is* sufficient for detection with a novel fluorescence MRI instrument when cells are placed in a ~40cm³ volume. Note that this signal is detected by a single photodiode - no camera is collecting spatial information (hence the measurement is independent of the direction of light arrival which would, for example, be impacted by scattering), and so FMT would not be possible. As such the benefit of MagLOV for this sensing is one of pure possibility – no standard fluorescence protein could have spatial information encoded in its fluorescence in the way that flavoproteins such as MagLOV can (and of these, MagLOV is unique in being a single protein unit amenable to directed evolution and exhibiting remarkably large MFE contrast).

3.13. Is it realistic to have magnetic field gradients that are sufficiently strong that MR contrast can be seen across the scale of a single cell and improve spatial resolution relative to fluorescence alone? The gradient across a 500 nm distance of >10 mT seems more than a bit challenging to achieve. Please comment.

In the updated manuscript and new applications demonstrated for spatial localisation (Fig 5 and Section 5.2) we now clarify that spatial localisation intended is the localisation of cells or fluorescent proteins in a larger volume, *not* the localisation of proteins within a cell. We did not pursue *that* as an immediate application because, as the reviewer suggests, it would be challenging to create appropriate field gradients. Nevertheless, having investigated this we do believe it may be possible and might be explored in future work. For example, cheap permanent magnets are available commercially with gradients of 200 mT/mm (link) which would give ~5µm resolution. In a more extreme example, gradients of up to 14 T/mm are produced in High-Gradient Magnetic Separators (link) (which admittedly is far from an MRI instrument, though our point is that such field gradients are not unimaginable – especially as the absolute field in our case is not ~T as it is for MRI). It is therefore not inconceivable that a setup could be constructed somewhere in between this range to deliver appropriate field gradients for sub-cellular localisation should the motivation exist. Nevertheless, we do not discuss or explore this possibility in the manuscript and will leave it for later adopters of our techniques to investigate should it benefit their applications.

3.13. These are interesting ideas, but if they are not practical, then it is unclear what the innovation is that warrants publication in Nature at this time.

As above, we hope the reviewer is pleased with the significant additions we have made throughout the manuscript – and especially direct demonstration of several of the applications we had initially hypothesised.

3.14. P. 5: specifically figure 2: Overall, experimental details are lacking - detection volumes, excitation intensities, imaging parameters, etc.

These details have now generally been included, please see SI. Shot-Noise Sensitivity for summary details.

3.15. I'll believe that these are hyperfine peaks, which is pretty cool, but the sampling seems pretty close to the limit for resolving these, and the data is quite noisy. This looks promising, but do the sidebands change frequency with applied magnetic field? or is the s/n too low and sampling time insufficient to determine this?

Following the reviewer's suggestion, we undertook a detailed analysis of our experimental setup design and data processing pipeline to see how we could validate or improve this measurement. Upon this investigation we conclude that we are not in fact *currently* able to resolve hyperfine peaks with appropriate signal-to-noise for them to be reported. The secondary peaks present in this data were an artefact due to the antenna design – which have similarly been observed and discussed in the context of hyperfine peaks in recent pre-prints from other groups (e.g. Fig 1F and Fig2A in [1]). Resolving this, in the new Supplementary Section S1.13.1 we explain how we now measure and calibrate-out non-uniformities in RF power to allow the new improved ODMR data in Fig1E.

[1] <https://www.biorxiv.org/content/10.1101/2025.04.16.649006v1>

3.16. The authors indicate that they need negative control cells to subtract signal off. So, an internal reference is needed as with G. Marriott's Optical lock-in detection (OLID) approach to fluorescence-based recovery? If so, this is more complicated than not needing an additional reference that must be subtracted. It is claimed that this is due to instrumental artefact/issues so without optical modulation it may be difficult to get rid of this modulated background. 2D suggests that other proteins/background emitters or species only give a small non-zero Delta I/I signal, or is this zero consistent with noise? Straight modulation of signal by varying the magnetic field should be able to give direct lock in detection without a negative control as the reference. Is there a path to fix this or will this always be an unavoidable problem?

How far was the magnet z-position moved to change the fields? what precision is needed for improving resolution or other applications? is this achievable?

There are several factors in this comment which we address step by step.

First, as shown in our updated data in Fig 1 we do *not* need negative control cells to subtract signal off. This was previously done only for the cellular conglomerate data (now Supplementary Figure S4) due to the large autofluorescence in that sample. However it is not done for any of the data in the manuscript's main text – i.e. all of the MFE/ODMR data does *not* require a negative control of non-magnetic-responsive to be subtracted. We clarify this in the updated manuscript with the move of the conglomerate data to the supplementary (as discussed in response to earlier reviewer points), and further explanation of our experimental method and data processing (in updated Supplementary Section S1.15).

Second, our data does *not* support that other proteins or background emitters give any Delta I/I signal. As the reviewer states, the data in that particular figure is consistent with noise. As further evidence for this, other measurements in the paper (e.g. of GFP emitting cells in lower panel of Fig 4F) show no magnetic field effect.

Third, for situations in which we need to *change* magnetic field (i.e. during a MFE experiments) we use an electromagnet (rather than permanent magnet) hence no movement is undertaken as field is controlled electronically. For ODMR experiments in the microscopy data (e.g. Fig 1E) we *do* move a permanent magnet between samples to achieve different field strengths; the amount of movement is determined by the magnetic field gradient which we measure as 1mT/mm (this is now discussed where this data is presented) meaning the span of data in Fig 1E involved a total magnet movement across a ~4.5mm range. For the *new* fluorescence MRI data (Fig 5) we once again have a static permanent magnet but now add a secondary helmholtz coil system to allow local magnetic field to be varied – and we show this volumetric localisation method can achieve a spatial resolution of on the order ~0.5mm. This could likely be improved with further development of the detection instrument, for example better optics and electronics for sensing, or by increasing the magnetic field gradient within the MRI volume to achieve narrower peaks in e.g. Fig 5E,F.

Finally, we have added citation to the Marriott 2008 paper on Lock-in detection, both as a motivator of the need for improved lock-in methodologies in our paper's updated Introduction, and as further quantitative comparison in the new Supplementary Section S1.22 that explains the utility and possible implementation of lock-in measurements with our approach.

3.17 Fig 2A: an entire continuous carpet of cells is detected here. what is the spot size and estimated number of cells for each of the measurements in panel 2E? what is the integration time for the data presented in B, C, E?

Several updates have been made to the manuscript to address this query, many of which are discussed in relation to points above. In summary:

- The data formerly in Fig2A is now moved to supplementary where it is used solely as an additional control experiment,
- Main text data showing ODMR (e.g. new Fig1D,E) is now that collected for single cells, and we state this is an average over $\sim 1,000$ cells; we provide both a single cell trace, this average over 1000 cells, and standard deviation *between* cells, which together we believe better explains the signal strength/variability of signal.
- Imaging integration time and acquisition protocols are clarified in the updated Supplementary Section S1.7 and e.g. Fig S2. For example, for all widefield single cell resolved measurements, exposure time was 200 ms per point. For MFE measurements imaging during magnet on/off periods allows collection of the dynamic response waveforms, and these are averaged over five on-off periods of 20 seconds each to give the resultant data (e.g. as in main text Fig 2). This process of image collection, processing, and subsequent averaging over multiple measurement periods is now presented in detail in Supplementary Section S1.15.2 and shown in new Fig. S10.

3.18. To improve resolution, you'd need a field gradient of $\sim 5\text{mT}/500\text{nm}$, correct (does this really mean a 10,000 T field needs to be applied for improved spatial resolution)? Is this in any way achievable or are my estimates from data presented incorrect?

For subcellular resolution, indeed very large *gradients* would be required, and this is discussed in response to point 3.13 above. This would not require large *absolute* fields however, so this is not completely infeasible (e.g. the absolute field could range from say 10-15 mT over a cell). Nevertheless, the generation of such fields is very speculative, and we restrict the spatial sensing applications we discuss to in-tissue type applications (i.e. locating cells, rather than separating locations *within* cells), where more reasonable gradients such as the 1 mT/mm we get from a simple permanent magnet, or 10 mT/mm as might be achieved with a more engineered approach could be realistically achieved to provide sub-mm accuracy.

3.19. I'd really like to see some actual application or a realistic proposed application based on data presented, not just saying we have a quantum sensor. Don't get me wrong, the data and innovation is quite cool, but it is unclear how this actually leads to anything biologically meaningful. I'm not saying it won't, I'm just not convinced by the hypothetical arguments the authors make based on the data presented.

As detailed in response to proofs above, the revised manuscript has been significantly expanded with novel applications, specifically spatial localisation in Fig 5 and sensing of local microenvironment in Fig 6. We have also added significant quantitative analysis of sensitivity and methodologies such as lock-in which we believe better places our

measurements in the context of the wider field, and will make it easier for other researchers to understand the properties/performance of MagLOV and apply it in their own broader applications.

3.20. P. 6: The description of where signal comes from (molecularly and quantum mechanically from the triplet states and their connections with RF and optical excitation is excellent.

We thank the reviewer for the positive feedback on this section. We have now added to it the following experiments (i) UV-Vis absorption spectroscopy to characterise the photoproduct under blue light illumination in purified MagLOV 2 R11f [Figures 3 D and S11 D], (ii) Fluorescence-detected MARY, i.e. MFE as a function of applied B [Fig S12], (iii) fluorescence lifetime and quantum yield estimation [Fig S13]. To further improve this, and in response to other reviewer comments, we have also added a new Supplementary Section S1.18 providing Spin Dynamics Simulations that connects this explanation with the fundamental processes and parameterises to the extent possible with the available data.

3.21. In contrast to the later statement of an artefact from the experimental system, this description suggests that no internal reference is needed and modulation of the magnetic field results in modulation of fluorescence signal as in fig 1.

Only later do the authors say this is because of the instrumental issues. this is important, as background subtraction using negative control cells to see signals in 2E is at best inconvenient, and at worst disqualifying for many applications. Can modulation with lock-in detection not automatically remove this unmodulated background? or does residual background modulation reduce contrast?

We have endeavoured to address many of these points above in the response to query 3.16 - in short, our measurements do *not* require background subtraction of the type suggested. We apologise that the measurement scheme was poorly explained in the original manuscript, leading to understandable confusion about how the measurements are attained.

- Fig 2E has been removed entirely, as we have come to understand that the noise present in those images was due to poor antenna design (this has been addressed, see the new Supplementary Section S1.13)
- We have included a new figure in the SI [Fig. S10] that illustrates what we mean by “background subtraction” for both MFE and OMR data, which is that due to drift in the long-term fluorescence level i.e. it is *not* a background signal present sample vs non-sample areas in a single image (such as that in Fig S4B). The background is not measured using negative controls or separately to the main signal, it is simply a fit based on the response trend expected if there were no MFE signal. We feel this is justified; for MFE it has been the standard

methodology for background subtraction for decades, for ODMR because in Fig. 1E the same background subtraction is applied to all sweeps and clearly the signal is not some artefact of the background subtraction (despite the background subtraction indeed having the inbuilt assumption that the resonance peak should occur between the start and end range of the frequency scan). This approach has been previously utilised in e.g. (Dejean, 2020) [1].

- Modulation with lock-in detection indeed can be used to assist with noise rejection, we demonstrate this in Fig. 4F and also use lock-in in both Fig 5E, F (1D-MRI) and Fig. 6 (Gadobutrol sensing).

[1] <https://pubs.rsc.org/en/content/articlelanding/2020/sc/d0sc01986k>

3.22. Are there specific parameters of the excitation or applied magnetic fields that could enable your label to be recovered without any background interference other than shot noise?

If the frequency spectrum of noise is known, it may be possible to tune the excitation, B0 field strength, and modulation rate to reject as much noise as possible. We expect in a typical scattering environment the noise is largely DC (scattered light and/or autofluorescence) and white (camera noise) which is rejected by any modulation frequency. Pink noise or 1/f noise may also be present which would require further consideration. In the new Supplementary Section S1.22 “Advantages of Lock-In Detection” we discuss a modulation rate of 3 Hz as a trade-off between response magnitude and number of lock-in cycles achieved in a given time. Recent work by Xiang et al. (2025) suggests modulation rates up to 10s of kHz may be possible, albeit with a significantly different experimental system and approach. Finally, and as we now state in the updated Discussion, further optimisation of MagLOV response magnitude and saturation time would be a good *general* approach to improve sensitivity of these measurements and would be a promising direction for future (bio)engineering work.

3.23. P. 7: only here is the instrumental artefact mentioned that needs to be subtracted. I appreciate the transparency, but this should be earlier as the reader is likely to be confused already about the need for negative control background subtraction.

Please see responses to points 3.16 and 3.21 above; this was only done in the case of the bulk sample, for which (a) we have moved the raw data (formerly Fig 2B of the paper) to SI Fig S4, and (b) removed the subtracted data plot (formerly Fig2C) as it does not add anything given the resulting signal still exhibited antenna induced artefacts and also suggests all measurements required negative control background subtraction - which is not the case and now clarified.

3.24. Further, in Fig. 2: the subtraction to yield hyperfine peak contrast from 2B to 2C is suggestive, but the noise is very high. The hyperfine side bands should not be present in the instrument artefact, correct? can you use a different modulation scheme (frequency modulation/heterodyne detection) to more directly recover your signal of interest without requiring a negative control cell population for subtraction?

Please see response to 3.15 above; regarding hyperfine peaks which we do not believe we have sufficient data to support. Nevertheless, more advanced modulation lock-in detection schemes (such as suggested by Xiang et al. 2025) are of significant interest but will require investigation.

3.25. P. 8: Characterization of species involved is excellent. Poor S/N and vague (and seemingly unrealistic) discussion of sensing applications, however, lessen impact as presented. The science and system are exciting though.

We appreciate the reviewer's enthusiasm for the detailed characterisation of our samples in Figure 3. This data is now expanded with additional measurements and supporting theory as explained in response to point 3.20. Regarding the S/N we appreciate this is limited in some cases, which we believe is an unavoidable consequence of the scattering nature of the bulk cell suspension (and inherent to any spectroscopic measurement with reduced signal integration as a result of dividing into wavelength bins). The S/N can be improved by measuring the purified MagLOV solution which we now present in Fig S13. However, we think the spectroscopic cell suspension measurements, despite their poor S/N, are nevertheless important, as they demonstrate that no 'confounding' species are present in the cell in comparison to purified protein solution.

We have now replaced Fig 3A with cell suspension spectra recorded with an Edinburgh Instruments spectrofluorimeter (details in the SI, same instrument as for panel C), and applied a moving average filter (window length = 3, i.e. averaging over the LHS and RHS neighbouring points). While the S/N on this spectrometer is higher and has an improved correction for wavelength-dependent PMT sensitivity (cf. to CCD employed in wavelength-resolved MFE set up), it suffers from laser excitation artefacts (spikes at 550 nm and 650 nm), which tend to appear when the count rate is relatively low.

Fig 3B now features the unnormalized ΔI (MFE) spectra, which have been smoothed with a moving average filter. Note that the spectral shapes in Panel A and B now differ slightly due to different illumination conditions and detectors employed. We previously opted to present this data normalised and without smoothing, however, the purpose of those plots is simply to show the entire spectrum exhibits the MFE. We hope that the additional processing makes the data easier to interpret for this purpose and addresses the reviewer's comment about S/N.

The vibrational fine structure evident in panel Fig 3C is crucial to infer the presence of a bound species in the bulk cell suspension. A moving average smoothing has now also been applied to this data to better resolve the main features from the noise. In excitation spectra, the S/N is also a trade-off between wavelength resolution and signal intensity (as the excitation power is controlled by the excitation bandwidth). To resolve the vibrational fine structure, we chose to sacrifice some S/N to obtain better wavelength resolution. However, we have now included UV-Vis absorption data of the purified MagLOV 2 fast solution (the new Fig 3D), which has a significantly better S/N, and clearly displays the same features as the bulk cell suspension data.

By demonstrating proof-of-concept of new sensing applications (e.g. Fig 5,6, discussed at length in response to points above), additional spectroscopic characterization of the purified MagLOV solution, and removing mention of more speculative sensing applications we hope to have improved the core message and address the reviewer's desire for discussion of applications to be as concrete as possible.

3.26. P. 9: the evolution/genetic engineering for MFE is also very nice.

We thank the reviewer for their enthusiasm about the directed evolution method presented.

3.27. P. 11: Fig 5 caption: Jablonski, et al as referenced above did this separation of two fluorescent proteins based on modulation. there are probably other examples of this from the same and other groups at this point. They used a second laser to depopulate the dark/triplet state, then could use lock-in amplification (i.e. a fourier transform) to decouple different species. EGFP, AcGFP.

We thank the reviewer for highlighting this past method; as in response to point 3.3 above discussion of this method is now included in the new Supplementary Section S1.22. Some further considerations:

- We believe our method for multiplexing of fluorescent reporters has a significantly different span of possible applications when compared to that cited from Jablonski. For example, ours does not require a complex optical setup with multiple lasers etc. As such both techniques represent multiplexing methods, but implemented with entirely different physical processes with their own strengths and weaknesses,
- At modulation frequencies below 100 Hz, the modulation depths of the Jablonski method [1] saturate at 7.5% in solution (12% immobilized). In our case, modulation depths of 30% and greater are possible albeit at very low modulation frequencies, and at 3 Hz similar modulation depths (12%) are achievable – see Fig. S15.

- Indeed, Orth et al [2] report an interesting multiplexing technique based on photobleaching characteristics. Multiplexing fluorescent reporters in general enables higher throughput screens and we see MagLOV as a good candidate for increasing the size of such screens by adding this dimension, which we now state explicitly in the manuscript at the end of Section 3.

[1] <https://pmc.ncbi.nlm.nih.gov/articles/PMC3570161/>

[2] <https://pmc.ncbi.nlm.nih.gov/articles/PMC6033574/>

3.28. in 5 B,C, Occurrences might be better for y-axis than frequency. frequency is used elsewhere so was initially confusing.

Thank you for this, as suggested we have changed the y axis to be in terms of counts.

3.29. Also, tau on and off are not explained in the caption at all, so figure does not stand on its own. Even in the text, are tau on/off defined? presumably the magnetic field is modulated and the rise and decay are measured or the period between B field on times is modulated to get these decay times? this needs to be described in more detail.

Again, apologies for this omission. In response we have first changed the fitting protocol to fit a *single* tau, as we believe the on/off rates should *in theory* follow the same rate parameter. We have also now updated the text, and the figure caption, to include the parameter tau, with the function fit explained in detail in Supplementary Section S1.15.2. This parameter has *also* now been calculated for the variant data in Fig 2 and bulk measurement data in Fig S11.

3.30. How are the two populations separated when mixed?

In this demonstration the populations are not *physically* separated post mixing. Previously, we simply chose a tau threshold based on the single variant data and applied it to the combined sample data. However, the single cell data can be noisy, occasionally leading to poor fitting. To make the classification more robust, we have therefore improved upon this by using a classification algorithm (XGBoost) trained on the single variant data to separately identify the variant response traces from the combined sample. The classification algorithm takes the entire (normalised) 1 period average trace as input. As such the data in Fig 4C shows two partially overlapping histograms; with the colour representing the predicted classification. We observe an 0.9:1 ratio of classifications (against an intended mixture of 1:1) supporting that the classifier is balanced, which alongside the clear bimodality supports that the method *is* able to distinguish multiple sub-populations – though we do not have a ground truth (like in Fig 4D) to compare against in this case. These points are explained in the updated manuscript, including discussion of reasons why the 0.9:1 ratio may differ from the ideal 1:1 for the combined sample.

As a broader response to this comment, if *physical* separation of different variants is desired it may be possible to achieve this post mixing via identifying them in measurements (e.g. using our demonstrated classification algorithm) and then using a digital micromirror device (DMD, as already integrated in our microscopy platform) to selectively target different populations with light, for instance killing one of the two sub-populations of cells using UV, then washing the mixture into media and growing to extract it. Similarly, such a screening method would enable much higher throughput selection of MagLOV and related protein variants than achieved with our current plate-based assays; development of such an approach will be a subject for future work.

3.31. Please provide experimental parameters (imaging time, modulation parameters, intensities, detection volumes, how demodulation is done, time traces vs frequency responses, etc)

As in response to related points above, we have significantly updated the methods described in the manuscript to make these aspects clear and reproducible. In particular, there is an updated Supplementary Section S1.7 which explains the imaging parameters; there is a new Supplementary Section S1.15 which sets out the modulation approach, data processing, and fitting protocols; methods for the volumetric measurements of bulk cell suspensions are also clarified in updated Section S1.17.

3.32. P. 13: what “quantum biological sensing”? robust quantum measurements? sure, spin is quantum, but so is fluorescence. I don’t see what is really sensed here as the s/n and resolution are both lacking on anything close to a single cell, let alone subcellular info.

With regards to quantum sensing, please see the response to reviewer comment 3.4. In short, we agree with the reviewer that there are multiple interpretations of the phrase “quantum sensing” and consequently we have made a more precise definition of this term in our paper’s introduction which we believe is in line with the standard terminology of the field. Further, we now show an additional quantum sensing application (Fig 6).

In addition to above, optimisation of our imaging approach has led to significant improvements in S/N throughout, and we additionally parameterise this e.g. with presentation of standard deviation in measurements across ~thousands of cells. Indeed, we do not imply we are sensing any *subcellular* structure or spatial information in the current work, and have clarified mentions of quantum sensing and spatial localisation to try to ensure our work is not interpreted this way.

3.33. The science overall is very nice, as is the characterization and evolution, but the “quantum sensing” are too far into the potential regime, and an actual sensing system should probably be shown or at least rationalized based on real data. Basically, the probe is being detected, with no real sensing occurring. Such sensing should improve

on what is currently possible (or show a path to doing so), without overselling the potential applications - especially when the sensitivity or suggested potential seems too low. I will be delighted to see data that shows otherwise, but I'm not sure the technology is there yet, or will ever be there (I hope I am wrong about this, but it is the authors' responsibility to demonstrate this).

As detailed in response to point 3.32 above we hope we have now generated proof-of-concept data which satisfies the requests of the reviewer. In particular, we have shown direct measurement of local spin environment with the new Fig 6; we have built a novel fluorescence MRI instrument and shown it can be used to localise MagLOV reporting cells in a large spatial volume; and in the new Supplementary Section S1.12 we have completed a quantitative analysis of our proteins' sensitivity and compare this with other methods used in the Quantum Sensing literature – while highlighting that our methods advantage as a *biologically encoded and controllable* sensor makes it uniquely suited to many applications where e.g. NV nanodiamonds are not suitable.

Further, we have edited the paper throughout to tightened up claims of application potentialities.

3.34. their imaging system generates (large) fields of ~ 1 T/m. not sufficient for intracellular or even cellular resolution based on ODMR. I don't have a good sense of what a realistic application would be that would advance the field - please explain if going beyond demonstration of ODMR.

As outlined in responses to points above, we hope the reviewer is now happy with our demonstration of localisation of MagLOV reporters within a 3D volume. As stated in the updated text, we believe this is a valuable application demonstration – for example, it has been implemented with an existing rodent MRI coil hence localisation of such a signal within a rodent model (e.g. for molecule localisation or tissue/specific gene expression study) would be a logical next step for future development. We also include multiple citations within the paper to other methods (significantly different to ours, with their own strengths and weaknesses) highlighting the general interest in such sensing methods.

3.35. tumbling rate for binding partners. maybe. but this would be for LOTS of proteins in solution due to sensitivity. this is ok, but not sure this can be done in cells or in real conditions. What are the timescales that can be probed and how does this compare with fluorescence anisotropy (and extensions of FA to larger complexes)? in vivo tumbling to measure interacting partners would be interesting, but again, need a very large sample, and fluorescence still must be detected (at 450 nm, no less, so in vivo applications are not viable).

We thank the author for their enthusiasm for this application, but appreciate their general desire for our manuscript to focus on applications which are concretely

supported by our work. As such we have removed mentions of tumbling rate measurement from the paper, however, we do now demonstrate a *different* molecular sensing application in Fig 6.

In summary, we have made significant changes throughout the manuscript to address the reviewer's concerns, and we hope they will be happy with the thoroughness with which these are addressed.

C) Response to Review 1

Referee #1 (Remarks to the Author):

The authors did a great job responding to reviewer comments. Congrats on a magnificent paper!

We thank all the reviewers for their thoughtful consideration and detailed critique which has led to a much-improved version of our paper.

D) Response to Reviewer 2

Referee #2 (Remarks to the Author):

1. The main change in the revised manuscript by Abrahams et al. is the inclusion of two new experimental datasets that demonstrate proof of concept for spatial localization and microenvironmental sensing. These additions address the common critique from all three referees regarding the lack of convincing application data.

We thank the reviewer for the earlier suggestion to include such data and are glad to hear the new experiments and data addresses the reviewer's critiques.

2. Spatial resolution in bulk cell samples is now demonstrated. However, the practical utility of this technique remains unclear. The reported spatial resolution is on the order of 5–10 mm, likely limited by both spin resonance linewidth and signal-to-noise ratio.

While magnetic fields can penetrate deep tissues, the fluorescence detection on which this method relies typically has a penetration depth of only a few millimeters, even under diffusive conditions.

We thank the reviewer for further discussion of this point. We would first like to state that the 5-10mm figure is not the resolution achieved in our work – it differs from our results (which are 0.5-1mm error) by an order of magnitude. For example, the calibrated estimation vs ground truth difference in Fig. 5F is $7.5-6.9 = 0.6\text{mm}$. Note that it may be the reviewer had interpreted this 5-10mm figure from the width of the peaks in Fig E,F. To clarify this previous point of uncertainty, in the updated Methods we now explain that the width of these peaks is directly determined by the ODMR linewidth (e.g. measured from Fig 1D,E) which in terms of lateral position is $\sim 3\text{mm}$ half-width in the Fig 5 instrument. However, given we *know* this linewidth (and it is a physical property of the MagLOV protein) it does *not* represent the limiting factor for resolution of our technique – as demonstrated by our single/dual sample measurements which are $\sim 0.5\text{mm}$ from ground truth.

To the reviewer's broader request for further explanation of the limitations and future developments of this technique, we have added discussion of this to Methods section, where we highlight straightforward improvements (many involving implementation of already-existing technologies) that could be made to the present instrument to achieve superior signal-to-noise.

3. The magnetic field sensing results in Fig. 6 confirm that the method is sensitive to the local microenvironment. The proof-of-concept results are interesting and may inspire readers to consider new directions for biomedical sensing. Nonetheless, the current sensitivity remains several orders of magnitude lower than that of NV centers.

Moreover, the reported integration time of 200-500 ms per point is relatively slow for in vivo sensing or imaging applications.

We appreciate the reviewer's drawing attention to this application, and we are similarly hopeful that our proof-of-concept results will inspire readers to further develop measurement techniques and the protein itself such that the sensitivity might be biologically useful. Thanks to the pre-print readership we are already aware of several groups actively engaged in these efforts. We would also highlight – as stated in our manuscript – that a major advantage of our method compared to NV centres is that it is biologically expressible, mitigating the significant and highly limiting requirement that NV present in their delivery to biological sites/applications of interest.

4. Since this revised version was submitted, Reference [50] by Feder et al. has been published in Nature (<https://www.nature.com/articles/s41586-025-09417-w>). Given the relevance in principle and scope, it would be appropriate to cite this paper in the Introduction, revise the discussion accordingly, and briefly compare the two techniques--particularly in terms of signal-to-noise ratio and measurement speed.

We agree that the work of Feder et al. is highly relevant; we have been in communication with the authors since prior to submitting our respective papers, and we had indeed previously cited their preprint in our manuscript and its supplementary notes. All citations are now updated to reflect the recent paper (rather than preprint), and we additionally mention it in the Introduction of our manuscript.

Following a recent conversation with their team held just after their work was published, we feel the best comparison figure of merit is the magnetic field sensitivity, which is the standard used across quantum sensing to compare different techniques. This metric has the benefit of being less dependent on specific details of the mechanism/measurement protocol compared to other measures. Consequently, we have made additions to Supplementary Note 1 (referred to in the main text) where we discuss this work and highlight that our method, when characterised with this metric, is significantly more sensitive. Furthermore, we *also* compare our work against Feder *et al* in terms of ODMR contrast and linewidth in the updated Table S2.

5. According to the author guidelines, the first paragraph should contain references (as in the original submission). The citation format of Reference 9 is also incomplete.

As requested by the editors and reviewer, references have been added to the abstract and the Hayward et al. citation has been corrected.

6. In Fig. 5B, please display the grayscale map.

We have added a grayscale colorbar/map as requested.

7. The methods used to obtain Figs. 5E and 5F are not clearly explained. Was the z-position determined by the frequency of the B_1 field? If so, it would be clearer to plot the data with frequency as the primary x-axis and the corresponding z-coordinate as a secondary (top) x-axis.

We thank the reviewer for the suggestion and apologize for not explaining with enough clarity. The z-position was determined by the net B_0 field offset (which is added to the gradient B_0 field) at which a resonance was detected with the 500 MHz B_1 field. I.e. the B_1 frequency does *not* change during this measurement. This is now clarified in the new Methods section of the manuscript.

Noting that the field gradient is ~ 1 (0.95 mT/mm) we are not sure that adding a secondary axis with *almost* the same values, but differing units would add clarity and therefore have not made this change. As the key purpose (and metric of interest) is locating a sample in *space*, and (in our opinion) this is what will be of most interest to general readers, we have elected to leave this axis in terms of distance (mm) rather than magnetic field strength.

Nevertheless, we are willing to include this additional axis should the reviewer or editor still feel this change would benefit the reader given the above.

8. The simultaneous measurements in Fig. 5F appear to exhibit higher noise (or fluctuations) and greater crosstalk between the two samples; please comment on it.

These fluorescence measurements were taken with a single photodiode with several optical filters (e.g. as in Fig 5C). As such this measurement collects the total fluorescence of both samples, hence the overall contrast per sample is lower as the background is doubled (which is also a contribution to this increased noise).

In general this instrument – developed here during the review process of this paper – is intended for proof of concept and could be improved significantly in the future. To capture this and inspire future work we have added a paragraph to the Methods section highlighting key limitations and future ways to overcome these. As an example, fast optical lock-in (additionally to the B1 on/off lock-in) and improved B0 field uniformity could both contribute to superior localisation accuracy and reduction in measurement noise.

9. There is also little description of the deconvolution method applied in Figs. 5E and 5F. What was the functional form of the point spread function? It is somewhat surprising that a simple deconvolution produced two clearly separated peaks from such noisy data.

As requested, we now outline the specific algorithms and settings used for the deconvolution in the updated Methods section. In particular, Lucy Richard deconvolution was used with a Gaussian impulse function of width 3mm. Note this 3mm value (as described in response to point 4 above) is estimated from the earlier ODMR measurements of MagLOV. We make this clear in the Methods, and that *other* properties of the data (e.g. the number/location of the peaks) were *not* provided to the deconvolution and as such this data processing methods represents one that could be used in practical application (e.g. it does not assume any unknown parameters of the sample prior to the measurement!)

10. In Fig. 6A, is the solid curve a theoretical fit using the function $(ax + b)^{-1} + c$? Please specify the best-fit parameters and compare them with theoretical expectations.

As requested, in the process of consolidating our paper's *Methods* we now describe the motivation for the specific theoretical fit in the last paragraph of the Microenvironment Sensing Methods. We also note that, while the order of magnitude for some parameters could be estimated from theory, this is not viable without measurements of T1/T2 times – developing experimental apparatus for this suitable to MagLOV and similar proteins will be a subject of future work in the field.